# Efficient and Modular Implicit Differentiation

**Mathieu Blondel, Quentin Berthet, Marco Cuturi**,[*] **Roy Frostig,**
**Stephan Hoyer, Felipe Llinares-López, Fabian Pedregosa, Jean-Philippe Vert**[*]
Google Research

## Abstract

Automatic differentiation (autodiff) has revolutionized machine learning. It allows to express complex computations by composing elementary ones in creative ways and removes the burden of computing their derivatives by hand. More recently, differentiation of optimization problem solutions has attracted widespread attention with applications such as optimization layers, and in bi-level problems such as hyper-parameter optimization and meta-learning. However, so far, implicit differentiation remained difficult to use for practitioners, as it often required case-by-case tedious mathematical derivations and implementations. In this paper, we propose automatic implicit differentiation, an efficient and modular approach for implicit differentiation of optimization problems. In our approach, the user defines directly in Python a function $F$ capturing the optimality conditions of the problem to be differentiated. Once this is done, we leverage autodiff of $F$ and the implicit function theorem to automatically differentiate the optimization problem. Our approach thus combines the benefits of implicit differentiation and autodiff. It is efficient as it can be added on top of any state-of-the-art solver and modular as the optimality condition specification is decoupled from the implicit differentiation mechanism. We show that seemingly simple principles allow to recover many existing implicit differentiation methods and create new ones easily. We demonstrate the ease of formulating and solving bi-level optimization problems using our framework. We also showcase an application to the sensitivity analysis of molecular dynamics.

## 1 Introduction

Automatic differentiation (autodiff) is now an inherent part of machine learning software. It allows to express complex computations by composing elementary ones in creative ways and removes the tedious burden of computing their derivatives by hand. In parallel, the differentiation of optimization problem solutions has found many applications. A classical example is bi-level optimization, which typically involves computing the derivatives of a nested optimization problem in order to solve an outer one. Examples of applications in machine learning include hyper-parameter optimization [23, 77, 70, 38, 12, 13], neural networks [59], and meta-learning [39, 72]. Another line of active research involving differentiation of optimization problem solutions are optimization layers [55, 6, 65, 31, 46], which can be used to encourage structured outputs, and implicit deep networks [8, 36, 43, 44, 73], which have a smaller memory footprint than backprop-trained networks.

Since optimization problem solutions typically do not enjoy an explicit formula in terms of their inputs, autodiff cannot be used directly to differentiate these functions. In recent years, two main approaches have been developed to circumvent this problem. The first one consists of unrolling the iterations of an optimization algorithm and using the final iteration as a proxy for the optimization problem solution [83, 32, 29, 39, 1]. This allows to **explicitly** construct a computational graph relating the algorithm output to the inputs, on which autodiff can then be used transparently. However, this requires a reimplementation of the algorithm using the autodiff system, and not all algorithms

---

[*]Work done while at Google Research, now at Apple and Owkin, respectively.

36th Conference on Neural Information Processing Systems (NeurIPS 2022).

are necessarily autodiff friendly. Moreover, forward-mode autodiff has time complexity that scales linearly with the number of variables and reverse-mode autodiff has memory complexity that scales linearly with the number of algorithm iterations. In contrast, a second approach consists in **implicitly** relating an optimization problem solution to its inputs using optimality conditions. In a machine learning context, such implicit differentiation has been used for stationarity conditions [11, 59], KKT conditions [23, 45, 6, 67, 66] and the proximal gradient fixed point [65, 12, 13]. An advantage of implicit differentiation is that a solver reimplementation is not needed, allowing to build upon decades of state-of-the-art software. Although implicit differentiation has a long history in numerical analysis [48, 10, 57, 20], so far, it remained difficult to use for practitioners, as it required a case-by-case tedious mathematical derivation and implementation. CasADi [7] allows to differentiate various optimization and root finding problem algorithms provided by the library. However, it does not allow to easily add implicit differentiation on top of existing solvers from optimality conditions expressed by the user, as we do. A recent tutorial explains how to implement implicit differentiation in JAX [34]. However, the tutorial requires the user to take care of low-level technical details and does not cover a large catalog of optimality condition mappings as we do. Other work [2] attempts to address this issue by adding implicit differentiation on top of cvxpy [30]. This works by reducing all convex optimization problems to a conic program and using conic programming's optimality conditions to derive an implicit differentiation formula. While this approach is very generic, solving a convex optimization problem using a conic programming solver—an ADMM-based splitting conic solver [68] in the case of cvxpy—is rarely state-of-the-art for every problem instance.

In this work, we ambition to achieve for optimization problem solutions what autodiff did for computational graphs. We propose **automatic implicit differentiation**, a simple approach to add implicit differentiation on top of any existing solver. In this approach, the user defines directly in Python a mapping function $F$ capturing the optimality conditions of the problem solved by the algorithm. Once this is done, we leverage autodiff of $F$ combined with the implicit function theorem to automatically differentiate the optimization problem solution. Our approach is **generic**, yet it can exploit the **efficiency** of state-of-the-art solvers. It therefore combines the benefits of implicit differentiation and autodiff. To summarize, we make the following contributions.

- We describe our framework and its JAX [21, 42] implementation (https://github.com/google/jaxopt/). Our framework significantly **lowers the barrier** to use implicit differentiation, thanks to the seamless integration in JAX, with low-level details all abstracted away.

- We instantiate our framework on a **large catalog** of optimality conditions (Table 1), recovering existing schemes and obtaining new ones, such as the mirror descent fixed point based one.

- On the theoretical side, we provide new bounds on the **Jacobian error** when the optimization problem is only solved approximately, and empirically validate them.

- We implement four **illustrative applications**, demonstrating our framework's ease of use.

Beyond our software implementation in JAX, we hope this paper provides a **self-contained blueprint** for creating an efficient and modular implementation of implicit differentiation in other frameworks.

**Notation.** We denote the gradient and Hessian of $f\colon \mathbb{R}^d \to \mathbb{R}$ evaluated at $x \in \mathbb{R}^d$ by $\nabla f(x) \in \mathbb{R}^d$ and $\nabla^2 f(x) \in \mathbb{R}^{d \times d}$. We denote the Jacobian of $F\colon \mathbb{R}^d \to \mathbb{R}^p$ evaluated at $x \in \mathbb{R}^d$ by $\partial F(x) \in \mathbb{R}^{p \times d}$. When $f$ or $F$ have several arguments, we denote the gradient, Hessian and Jacobian in the $i^{\text{th}}$ argument by $\nabla_i$, $\nabla_i^2$ and $\partial_i$, respectively. The standard probability simplex is denoted by $\triangle^d := \{x \in \mathbb{R}^d \colon \|x\|_1 = 1, x \geq 0\}$. For any set $\mathcal{C} \subset \mathbb{R}^d$, we denote the indicator function $I_{\mathcal{C}}\colon \mathbb{R}^d \to \mathbb{R} \cup \{+\infty\}$ where $I_{\mathcal{C}}(x) = 0$ if $x \in \mathcal{C}$, $I_{\mathcal{C}}(x) = +\infty$ otherwise. For a vector or matrix $A$, we note $\|A\|$ the Frobenius (or Euclidean) norm, and $\|A\|_{\text{op}}$ the operator norm.

## 2 Automatic implicit differentiation

### 2.1 General principles

**Overview.** Contrary to autodiff through unrolled algorithm iterations, implicit differentiation typically involves a manual, sometimes complicated, mathematical derivation. For instance, numerous works [23, 45, 6, 67, 66] use Karush–Kuhn–Tucker (KKT) conditions in order to relate a constrained optimization problem's solution to its inputs, and to manually derive a formula for its derivatives. The derivation and implementation in these works are typically case-by-case.

```
X_train, y_train = load_data()   # Load features and labels

def f(x, theta):  # Objective function
  residual = jnp.dot(X_train, x) - y_train
  return (jnp.sum(residual ** 2) + theta * jnp.sum(x ** 2)) / 2

# Since f is differentiable and unconstrained, the optimality
# condition F is simply the gradient of f in the 1st argument
F = jax.grad(f, argnums=0)

@custom_root(F)
def ridge_solver(init_x, theta):
  del init_x  # Initialization not used in this solver
  XX = jnp.dot(X_train.T, X_train)
  Xy = jnp.dot(X_train.T, y_train)
  I = jnp.eye(X_train.shape[1])  # Identity matrix
  # Finds the ridge reg solution by solving a linear system
  return jnp.linalg.solve(XX + theta * I, Xy)

init_x = None
print(jax.jacobian(ridge_solver, argnums=1)(init_x, 10.0))
```

Figure 1: Adding implicit differentiation on top of a ridge regression solver. The function $f(x, \theta)$ defines the objective function and the mapping $F$, here simply equation (4), captures the optimality conditions. Our decorator `@custom_root` automatically adds implicit differentiation to the solver for the user, overriding JAX's default behavior. The last line evaluates the Jacobian at $\theta = 10$.

In this work, we propose a generic way to easily add implicit differentiation on top of existing solvers. In our approach, the user defines directly in Python a mapping function $F$ capturing the optimality conditions of the problem solved by the algorithm. We provide reusable building blocks to easily express such $F$. The provided $F$ is then plugged into our Python decorator `@custom_root`, which we append on top of the solver declaration we wish to differentiate. Under the hood, we combine the implicit function theorem and autodiff of $F$ to automatically differentiate the optimization problem solution. A simple illustrative example is given in Figure 1.

**Differentiating a root.** Let $F: \mathbb{R}^d \times \mathbb{R}^n \to \mathbb{R}^d$ be a user-provided mapping, capturing the optimality conditions of a problem. An optimal solution, denoted $x^\star(\theta)$, should be a **root** of $F$:

$$F(x^\star(\theta), \theta) = 0 . \tag{1}$$

We can see $x^\star(\theta)$ as an implicitly defined function of $\theta \in \mathbb{R}^n$, i.e., $x^\star: \mathbb{R}^n \to \mathbb{R}^d$. More precisely, from the **implicit function theorem** [48, 57], we know that for $(x_0, \theta_0)$ satisfying $F(x_0, \theta_0) = 0$ with a continuously differentiable $F$, if the Jacobian $\partial_1 F$ evaluated at $(x_0, \theta_0)$ is a square invertible matrix, then there exists a function $x^\star(\cdot)$ defined on a neighborhood of $\theta_0$ such that $x^\star(\theta_0) = x_0$. Furthermore, for all $\theta$ in this neighborhood, we have that $F(x^\star(\theta), \theta) = 0$ and $\partial x^\star(\theta)$ exists. Using the chain rule, the Jacobian $\partial x^\star(\theta)$ satisfies

$$\partial_1 F(x^\star(\theta), \theta) \partial x^\star(\theta) + \partial_2 F(x^\star(\theta), \theta) = 0 .$$

Computing $\partial x^\star(\theta)$ therefore boils down to the resolution of the linear system of equations

$$\underbrace{-\partial_1 F(x^\star(\theta), \theta)}_{A \in \mathbb{R}^{d \times d}} \underbrace{\partial x^\star(\theta)}_{J \in \mathbb{R}^{d \times n}} = \underbrace{\partial_2 F(x^\star(\theta), \theta)}_{B \in \mathbb{R}^{d \times n}} . \tag{2}$$

When (1) is a one-dimensional root finding problem ($d = 1$), (2) becomes particularly simple since we then have $\nabla x^\star(\theta) = B^\top / A$, where $A$ is a scalar value.

We will show that existing and new implicit differentiation methods all reduce to this simple principle. We call our approach **automatic implicit differentiation** as the user can freely express the optimization solution to be differentiated through the optimality conditions $F$. Our approach is **efficient** as it can be added on top of any state-of-the-art solver and **modular** as the optimality condition specification is **decoupled** from the implicit differentiation mechanism. This contrasts with existing works, where the derivation and implementation are specific to each optimality condition.

**Differentiating a fixed point.** We will encounter numerous applications where $x^\star(\theta)$ is instead implicitly defined through a **fixed point**:

$$x^\star(\theta) = T(x^\star(\theta), \theta) ,$$

where $T\colon \mathbb{R}^d \times \mathbb{R}^n \to \mathbb{R}^d$. This can be seen as a particular case of (1) by defining the **residual**

$$F(x, \theta) = T(x, \theta) - x \,. \tag{3}$$

In this case, when $T$ is continuously differentiable, using the chain rule, we have

$$A = -\partial_1 F(x^\star(\theta), \theta) = I - \partial_1 T(x^\star(\theta), \theta) \quad \text{and} \quad B = \partial_2 F(x^\star(\theta), \theta) = \partial_2 T(x^\star(\theta), \theta).$$

**Computing JVPs and VJPs.** In most practical scenarios, it is not necessary to explicitly form the Jacobian matrix, and instead it is sufficient to left-multiply or right-multiply by $\partial_1 F$ and $\partial_2 F$. These are called vector-Jacobian product (VJP) and Jacobian-vector product (JVP), and are useful for integrating $x^\star(\theta)$ with reverse-mode and forward-mode autodiff, respectively. Oftentimes, $F$ will be explicitly defined. In this case, computing the VJP or JVP can be done via autodiff. In some cases, $F$ may itself be implicitly defined, for instance when $F$ involves the solution of a variational problem. In this case, computing the VJP or JVP will itself involve implicit differentiation.

The right-multiplication (JVP) between $J = \partial x^\star(\theta)$ and a vector $v$, $Jv$, can be computed efficiently by solving $A(Jv) = Bv$. The left-multiplication (VJP) of $v^\top$ with $J$, $v^\top J$, can be computed by first solving $A^\top u = v$. Then, we can obtain $v^\top J$ by $v^\top J = u^\top A J = u^\top B$. Note that when $B$ changes but $A$ and $v$ remain the same, we do not need to solve $A^\top u = v$ once again. This allows to compute the VJP w.r.t. different variables while solving only one linear system.

To solve these linear systems, we can use the conjugate gradient method [51] when $A$ is symmetric positive semi-definite and GMRES [75] or BiCGSTAB [81] otherwise. These algorithms are all matrix-free: they only require matrix-vector products. Thus, all we need from $F$ is its JVPs or VJPs. An alternative to GMRES/BiCGSTAB is to solve the normal equation $AA^\top u = Av$ using conjugate gradient. This can be implemented using JAX's transpose routine `jax.linear_transpose` [41]. In case of non-invertibility, a common heuristic is to solve a least squares $\min_J \|AJ - B\|^2$ instead.

**Pre-processing and post-processing mappings.** Oftentimes, the goal is not to differentiate $\theta$ per se, but the parameters of a function producing $\theta$. One example of such pre-processing is to convert the parameters to be differentiated from one form to another canonical form, such as a quadratic program [6] or a conic program [2]. Another example is when $x^\star(\theta)$ is used as the output of a neural network layer, in which case $\theta$ is produced by the previous layer. Likewise, $x^\star(\theta)$ will often not be the final output we want to differentiate. One example of such post-processing is when $x^\star(\theta)$ is the solution of a dual program and we apply the dual-primal mapping to recover the solution of the primal program. Another example is the application of a loss function, in order to reduce $x^\star(\theta)$ to a scalar value. We leave the differentiation of such pre/post-processing mappings to the autodiff system, allowing to compose functions in complex ways.

**Implementation details.** When a solver function is decorated with `@custom_root`, we use `jax.custom_jvp` and `jax.custom_vjp` to automatically add custom JVP and VJP rules to the function, overriding JAX's default behavior. As mentioned above, we use linear system solvers based on matrix-vector products and therefore we only need access to $F$ through the JVP or VJP with $\partial_1 F$ and $\partial_2 F$. This is done by using `jax.jvp` and `jax.vjp`, respectively. Note that, as in Figure 1, the definition of $F$ will often include a gradient mapping $\nabla_1 f(x, \theta)$. Thankfully, JAX supports second-order derivatives transparently. For convenience, our library also provides a `@custom_fixed_point` decorator, for adding implicit differentiation on top of a solver, given a fixed point iteration $T$; see code examples in Appendix B.

## 2.2 Examples

We now give various examples of mapping $F$ or fixed point iteration $T$, recovering existing implicit differentiation methods and creating new ones. Each choice of $F$ or $T$ implies different trade-offs in terms of **computational oracles**; see Table 1. Source code examples are given in Appendix B.

**Stationary point condition.** The simplest example is to differentiate through the implicit function

$$x^\star(\theta) = \operatorname*{argmin}_{x \in \mathbb{R}^d} f(x, \theta),$$

where $f\colon \mathbb{R}^d \times \mathbb{R}^n \to \mathbb{R}$ is twice differentiable, $\nabla_1 f$ is continuously differentiable, and $\nabla_1^2 f$ is invertible at $(x^\star(\theta), \theta)$. In this case, $F$ is simply the gradient mapping

$$F(x, \theta) = \nabla_1 f(x, \theta). \tag{4}$$

Table 1: Summary of optimality condition mappings. Oracles are accessed through their JVP or VJP.

| Name | Equation | Solution needed | Oracle |
|---|---|---|---|
| Stationary | (4), (5) | Primal | $\nabla_1 f$ |
| KKT | (6) | Primal *and* dual | $\nabla_1 f, H, G, \partial_1 H, \partial_1 G$ |
| Proximal gradient | (7) | Primal | $\nabla_1 f, \text{prox}_{\eta g}$ |
| Projected gradient | (9) | Primal | $\nabla_1 f, \text{proj}_{\mathcal{C}}$ |
| Mirror descent | (13) | Primal | $\nabla_1 f, \text{proj}_{\mathcal{C}}^{\varphi}, \nabla\varphi$ |
| Newton | (14) | Primal | $[\nabla_1^2 f(x,\theta)]^{-1}, \nabla_1 f(x,\theta)$ |
| Block proximal gradient | (15) | Primal | $[\nabla_1 f]_j, [\text{prox}_{\eta g}]_j$ |
| Conic programming | (18) | Residual map root | $\text{proj}_{\mathbb{R}^p \times \mathcal{K}^* \times \mathbb{R}_+}$ |

We then have $\partial_1 F(x,\theta) = \nabla_1^2 f(x,\theta)$ and $\partial_2 F(x,\theta) = \partial_2 \nabla_1 f(x,\theta)$, the Hessian of $f$ in its first argument and the Jacobian in the second argument of $\nabla_1 f(x,\theta)$. In practice, we use autodiff to compute Jacobian products automatically. Equivalently, we can use the **gradient descent fixed point**

$$T(x,\theta) = x - \eta\nabla_1 f(x,\theta), \tag{5}$$

for all $\eta > 0$. Using (3), it is easy to check that we obtain the same linear system since $\eta$ cancels out.

**KKT conditions.** As a more advanced example, we now show that the KKT conditions, manually differentiated in several works [23, 45, 6, 67, 66], fit our framework. As we will see, the key will be to group the optimal primal and dual variables as our $x^\star(\theta)$. Let us consider the general problem

$$\underset{z \in \mathbb{R}^p}{\text{argmin}} \, f(z,\theta) \quad \text{subject to} \quad G(z,\theta) \leq 0, \, H(z,\theta) = 0,$$

where $z \in \mathbb{R}^p$ is the primal variable, $f\colon \mathbb{R}^p \times \mathbb{R}^n \to \mathbb{R}$, $G\colon \mathbb{R}^p \times \mathbb{R}^n \to \mathbb{R}^r$ and $H\colon \mathbb{R}^p \times \mathbb{R}^n \to \mathbb{R}^q$ are twice differentiable convex functions, and $\nabla_1 f$, $\partial_1 G$ and $\partial_1 H$ are continuously differentiable. The stationarity, primal feasibility and complementary slackness conditions give

$$\nabla_1 f(z,\theta) + [\partial_1 G(z,\theta)]^\top \lambda + [\partial_1 H(z,\theta)]^\top \nu = 0$$
$$H(z,\theta) = 0$$
$$\lambda \circ G(z,\theta) = 0, \tag{6}$$

where $\nu \in \mathbb{R}^q$ and $\lambda \in \mathbb{R}_+^r$ are the dual variables, also known as KKT multipliers. The primal and dual feasibility conditions can be ignored almost everywhere [34]. The system of (potentially nonlinear) equations (6) fits our framework, as we can group the primal and dual solutions as $x^\star(\theta) = (z^\star(\theta), \nu^\star(\theta), \lambda^\star(\theta))$ to form the root of a function $F(x^\star(\theta), \theta)$, where $F\colon \mathbb{R}^d \times \mathbb{R}^n \to \mathbb{R}^d$ and $d = p + q + r$. The primal and dual solutions can be obtained from a generic solver, such as an interior point method. In practice, the above mapping $F$ will be defined directly in Python (see Figure 7 in Appendix B) and $F$ will be differentiated automatically via autodiff.

**Proximal gradient fixed point.** Unfortunately, not all algorithms return both primal and dual solutions. Moreover, if the objective contains non-smooth terms, proximal gradient descent may be more efficient. We now discuss its fixed point [65, 12, 13]. Let $x^\star(\theta)$ be implicitly defined as

$$x^\star(\theta) := \underset{x \in \mathbb{R}^d}{\text{argmin}} \, f(x,\theta) + g(x,\theta),$$

where $f\colon \mathbb{R}^d \times \mathbb{R}^n \to \mathbb{R}$ is twice-differentiable convex and $g\colon \mathbb{R}^d \times \mathbb{R}^n \to \mathbb{R}$ is convex but possibly non-smooth. Let us define the proximity operator associated with $g$ by

$$\text{prox}_g(y,\theta) := \underset{x \in \mathbb{R}^d}{\text{argmin}} \, \frac{1}{2}\|x - y\|_2^2 + g(x,\theta).$$

To implicitly differentiate $x^\star(\theta)$, we use the fixed point mapping [69, p.150]

$$T(x,\theta) = \text{prox}_{\eta g}(x - \eta\nabla_1 f(x,\theta), \theta), \tag{7}$$

for any step size $\eta > 0$. The proximity operator is 1-Lipschitz continuous [64]. By Rademacher's theorem, it is differentiable almost everywhere. If, in addition, it is continuously differentiable in a neighborhood of $(x^\star(\theta), \theta)$ and if $I - \partial_1 T(x^\star(\theta), \theta)$ is invertible, then our framework to differentiate $x^\star(\theta)$ applies. Similar assumptions are made in [13]. Many proximity operators enjoy a closed form and can easily be differentiated, as discussed in Appendix C. An implementation is given in Figure 2.

```
grad = jax.grad(f)  # Pre-compile the gradient.

def T(x, theta):
    # Unpack the parameters of f and g.
    theta_f, theta_g = theta
    # Return the fixed point condition evaluated at x.
    return prox(x - grad(x, theta_f), theta_g)
```

Figure 2: Implementation of the proximal gradient fixed point (7) with step size $\eta = 1$.

**Projected gradient fixed point.** As a special case, when $g(x, \theta)$ is the indicator function $I_{\mathcal{C}(\theta)}(x)$, where $\mathcal{C}(\theta)$ is a convex set depending on $\theta$, we obtain

$$x^\star(\theta) = \underset{x \in \mathcal{C}(\theta)}{\operatorname{argmin}} f(x, \theta). \tag{8}$$

The proximity operator $\operatorname{prox}_g$ becomes the Euclidean projection onto $\mathcal{C}(\theta)$

$$\operatorname{prox}_g(y, \theta) = \operatorname{proj}_\mathcal{C}(y, \theta) := \underset{x \in \mathcal{C}(\theta)}{\operatorname{argmin}} \|x - y\|_2^2$$

and (7) becomes the projected gradient fixed point

$$T(x, \theta) = \operatorname{proj}_\mathcal{C}(x - \eta \nabla_1 f(x, \theta), \theta). \tag{9}$$

Compared to the KKT conditions, this fixed point is particularly suitable when the projection enjoys a closed form. We discuss how to compute the JVP / VJP for a wealth of convex sets in Appendix C.

**Current limitations.** While we have not observed issues in practice, we note that the approach developed in this section theoretically only applies to settings where the implicit function theorem is valid, namely, where optimality conditions satisfy the differentiability and invertibility conditions stated in §2.1. While this covers a wide range of situations even for non-smooth optimization problems (e.g., under mild assumptions the solution of a Lasso regression can be differentiated a.e. with respect to the regularization parameter, see Appendix E), an interesting direction for future work is to extend the framework to handle cases where the differentiability and invertibility conditions are not satisfied, using, e.g., the theory of nonsmooth implicit function theorems [25, 19].

## 3 Jacobian precision guarantees

In practice, either by the limitations of finite precision arithmetic or because we perform a finite number of iterations, we rarely reach the exact solution $x^\star(\theta)$. Instead, we reach an approximate solution $\hat{x}$ and apply the implicit differentiation equation (2) at this approximate solution. This motivates the need for precision guarantees of this approach. We introduce the following formalism.

**Definition 1.** *Let $F : \mathbb{R}^d \times \mathbb{R}^n \to \mathbb{R}^d$ be a continuously differentiable optimality criterion mapping. Let $A := -\partial_1 F$ and $B := \partial_2 F$. We define the **Jacobian estimate** at $(x, \theta)$, when $A(x, \theta)$ is invertible, as the solution to the linear equation $A(x, \theta)J(x, \theta) = B(x, \theta)$. It is a function $J : \mathbb{R}^d \times \mathbb{R}^n \to \mathbb{R}^{d \times n}$.*

It holds by construction that $J(x^\star(\theta), \theta) = \partial x^\star(\theta)$. Computing $J(\hat{x}, \theta)$ for an approximate solution $\hat{x}$ of $x^\star(\theta)$ therefore allows to approximate the true Jacobian $\partial x^\star(\theta)$. In practice, an algorithm used to solve (1) depends on $\theta$. Note however that, what we compute is not the Jacobian of $\hat{x}(\theta)$, unlike works differentiating through unrolled algorithm iterations, but an estimate of $\partial x^\star(\theta)$. We therefore use the notation $\hat{x}$, leaving the dependence on $\theta$ implicit.

We develop bounds of the form $\|J(\hat{x}, \theta) - \partial x^\star(\theta)\| < C\|\hat{x} - x^\star(\theta)\|$, hence showing that the error on the estimated Jacobian is at most of the same order as that of $\hat{x}$ as an approximation of $x^\star(\theta)$. These bounds are based on the following main theorem, whose proof is included in Appendix D.

**Theorem 1.** *Let $F : \mathbb{R}^d \times \mathbb{R}^n \to \mathbb{R}^d$ be continuously differentiable. If there are $\alpha, \beta, \gamma, \varepsilon, R > 0$ s.t. $A = -\partial_1 F$ and $B = \partial_2 F$ satisfy, for all $v \in \mathbb{R}^d$, $\theta \in \mathbb{R}^n$ and $x$ s.t. $\|x - x^\star(\theta)\| \le \varepsilon$:*

*A is well-conditioned, Lipschitz: $\|A(x, \theta)v\| \ge \alpha\|v\|$, $\|A(x, \theta) - A(x^\star(\theta), \theta)\|_{\operatorname{op}} \le \gamma\|x - x^\star(\theta)\|$.*

*B is bounded and Lipschitz: $\|B(x^\star(\theta), \theta)\| \le R$, $\|B(x, \theta) - B(x^\star(\theta), \theta)\| \le \beta\|x - x^\star(\theta)\|$.*

*Under these conditions, when $\|\hat{x} - x^\star(\theta)\| \leq \varepsilon$, we have*

$$\|J(\hat{x}, \theta) - \partial x^\star(\theta)\| \leq \left(\beta\alpha^{-1} + \gamma R\alpha^{-2}\right)\|\hat{x} - x^\star(\theta)\|.$$

This result is inspired by [52, Theorem 7.2], that is concerned with the stability of solutions to inverse problems. As a difference, we consider that $A(\cdot, \theta)$ is uniformly well-conditioned, rather than only at $x^\star(\theta)$. This does not affect the first order in $\varepsilon$ of this bound, and makes it valid for all $\hat{x}$. Our goal with Theorem 1 is to provide a result that works for general $F$ but can be tailored to specific cases.

In particular, for the gradient descent fixed point (5), this yields

$$A(x, \theta) = \eta\nabla_1^2 f(x, \theta) \text{ and } B(x, \theta) = -\eta\partial_2\nabla_1 f(x, \theta).$$

By specializing Theorem 1 for this fixed point, we obtain Jacobian precision guarantees with conditions directly on $f$ rather than $F$; see Corollary 1 in Appendix D. These guarantees hold for instance for the dataset distillation experiment in Section 4. Our analysis reveals in particular that Jacobian estimation by implicit differentiation **gains a factor of t compared to automatic differentiation**, after $t$ iterations of gradient descent in the strongly-convex setting [1, Proposition 3.2]. While our guarantees concern the Jacobian of $x^\star(\theta)$, we note that other studies [47, 54, 13] give guarantees on hypergradients (i.e., the gradient of an outer objective).

We illustrate these results on ridge regression, where $x^\star(\theta) = \text{argmin}_x \|\Phi x - y\|_2^2 + \sum_i \theta_i x_i^2$. This problem has the merit that the solution $x^\star(\theta)$ and its Jacobian $\partial x^\star(\theta)$ are available in closed form. By running gradient descent for $t$ iterations, we obtain an estimate $\hat{x}$ of $x^\star(\theta)$ and an estimate $J(\hat{x}, \theta)$ of $\partial x^\star(\theta)$; cf. Definition 1. By doing so for different numbers of iterations $t$, we can graph the relation between the error $\|x^\star(\theta) - \hat{x}\|_2$ and the error $\|\partial x^\star(\theta) - J(\hat{x}, \theta)\|_2$, as shown in Figure 3, empirically validating Theorem 1. The results in Figure 3 were obtained using the diabetes dataset from [35], with other datasets yielding a qualitatively similar behavior. We derive similar guarantees in Corollary 2 in Appendix D for proximal gradient descent.

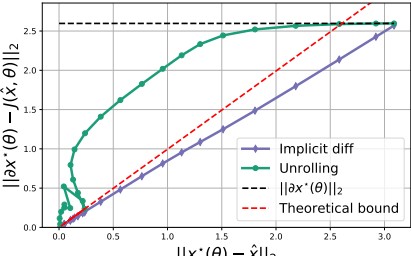

Figure 3: Jacobian estimate errors. Empirical error of implicit differentiation follows closely the theoretical upper bound. Unrolling achieves a much worse error for comparable iterate error.

## 4 Experiments

In this section, we demonstrate the ease of solving bi-level optimization problems with our framework. We also present an application to the sensitivity analysis of molecular dynamics.

### 4.1 Hyperparameter optimization of multiclass SVMs

In this example, we consider the hyperparameter optimization of multiclass SVMs [27] trained in the dual. Here, $x^\star(\theta)$ is the optimal dual solution, a matrix of shape $m \times k$, where $m$ is the number of training examples and $k$ is the number of classes, and $\theta \in \mathbb{R}_+$ is the regularization parameter. The challenge in differentiating $x^\star(\theta)$ is that each row of $x^\star(\theta)$ is constrained to belong to the probability simplex $\triangle^k$. More formally, let $X_{\text{tr}} \in \mathbb{R}^{m \times p}$ be the training feature matrix and $Y_{\text{tr}} \in \{0, 1\}^{m \times k}$ be the training labels (in row-wise one-hot encoding). Let $W(x, \theta) := X_{\text{tr}}^\top(Y_{\text{tr}} - x)/\theta \in \mathbb{R}^{p \times k}$ be the dual-primal mapping. Then, we consider the following bi-level optimization problem

$$\underbrace{\min_{\theta=\exp(\lambda)} \frac{1}{2}\|X_{\text{val}}W(x^\star(\theta), \theta) - Y_{\text{val}}\|_F^2}_{\text{outer problem}} \quad \text{subject to} \quad \underbrace{x^\star(\theta) = \underset{x\in\mathcal{C}}{\text{argmin}}\, f(x, \theta) := \frac{\theta}{2}\|W(x, \theta)\|_F^2 + \langle x, Y_{\text{tr}}\rangle,}_{\text{inner problem}}$$

where $\mathcal{C} = \triangle^k \times \cdots \times \triangle^k$ is the Cartesian product of $m$ probability simplices. We apply the change of variable $\theta = \exp(\lambda)$ in order to guarantee that the hyper-parameter $\theta$ is positive. The matrix $W(x^\star(\theta), \theta) \in \mathbb{R}^{p \times k}$ contains the optimal primal solution, the feature weights for each class. The outer loss is computed against validation data $X_{\text{val}}$ and $Y_{\text{val}}$.

While KKT conditions can be used to differentiate $x^\star(\theta)$, a more direct way is to use the projected gradient fixed point (9). The projection onto $\mathcal{C}$ can be easily computed by row-wise projections on the

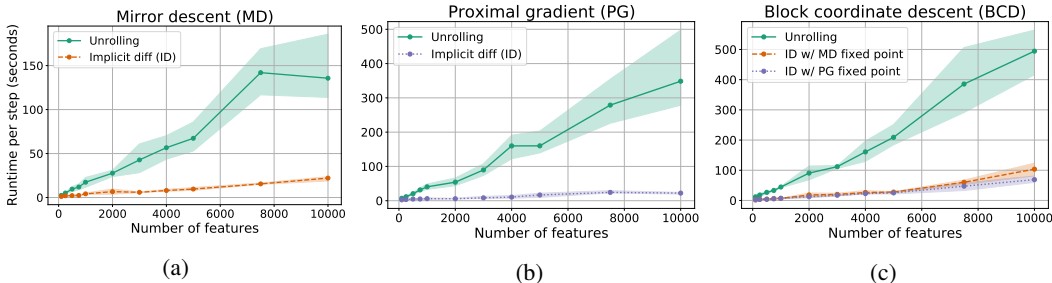

(a)          (b)          (c)

Figure 4: CPU runtime comparison of implicit differentiation and unrolling for hyperparameter optimization of multiclass SVMs for multiple problem sizes. Error bars represent 90% confidence intervals. **(a)** Mirror descent (MD) solver, with MD fixed point for differentiation. **(b)** Proximal gradient (PG) solver, with PG fixed point for differentiation. **(c)** Block coordinate descent solver; for implicit differentiation we obtain $x^\star(\theta)$ by BCD but perform differentiation with the MD and PG fixed points. This shows that the solver and fixed point can be independently chosen.

simplex. The projection's Jacobian enjoys a closed form (Appendix C). Another way to differentiate $x^\star(\theta)$ is using the mirror descent fixed point (13). Under the KL geometry, projections correspond to a row-wise softmax. They are therefore easy to compute and differentiate. Figure 4 compares the runtime performance of implicit differentiation vs. unrolling for the latter two fixed points.

## 4.2 Dataset distillation

Dataset distillation [82, 59] aims to learn a small synthetic training dataset such that a model trained on this learned data set achieves a small loss on the original training set. Formally, let $X_{\mathtt{tr}} \in \mathbb{R}^{m \times p}$ and $y_{\mathtt{tr}} \in [k]^m$ denote the original training set. The distilled dataset will contain one prototype example for each class and therefore $\theta \in \mathbb{R}^{k \times p}$. The dataset distillation problem can then naturally be cast as a bi-level problem, where in the inner problem we estimate a logistic regression model $x^\star(\theta) \in \mathbb{R}^{p \times k}$ trained on the distilled images $\theta \in \mathbb{R}^{k \times p}$, while in the outer problem we want to minimize the loss achieved by $x^\star(\theta)$ over the training set:

$$\underbrace{\min_{\theta \in \mathbb{R}^{k \times p}} f(x^\star(\theta), X_{\mathtt{tr}}; y_{\mathtt{tr}})}_{\text{outer problem}} \quad \text{subject to} \quad x^\star(\theta) \in \underbrace{\operatorname*{argmin}_{x \in \mathbb{R}^{p \times k}} f(x, \theta; [k]) + \varepsilon \|x\|^2}_{\text{inner problem}}, \quad (10)$$

where $f(W, X; y) := \ell(y, XW)$, $\ell$ denotes the multiclass logistic regression loss, and $\varepsilon = 10^{-3}$ is a regularization parameter that we found had a very positive effect on convergence.

In this problem, and unlike in the general hyperparameter optimization setup, *both* the inner and outer problems are high-dimensional, making it an ideal test-bed for gradient-based bi-level optimization methods. For this experiment, we use the MNIST dataset. The number of parameters in the inner problem is $p = 28^2 = 784$. while the number of parameters of the outer loss is $k \times p = 7840$. We solve this problem using gradient descent on both the inner and outer problem, with the gradient of the outer loss computed using implicit differentiation, as described in §2. This is fundamentally different from the approach used in the original paper, where they used differentiation of the unrolled iterates instead. For the same solver, we found that the implicit differentiation approach was 4 times faster than the original one. The obtained distilled images $\theta$ are visualized in Figure 5.



Figure 5: Distilled dataset $\theta \in \mathbb{R}^{k \times p}$ obtained by solving (10).

## 4.3 Task-driven dictionary learning

Task-driven dictionary learning was proposed to learn sparse codes for input data in such a way that the codes solve an outer learning problem [60, 78, 85]. Formally, given a data matrix $X_{\mathtt{tr}} \in \mathbb{R}^{m \times p}$

Table 2: Mean AUC (and 95% confidence interval) for the cancer survival prediction problem.

| Method | $L_1$ logreg | $L_2$ logreg | DictL + $L_2$ logreg | Task-driven DictL |
|---|---|---|---|---|
| AUC (%) | $71.6 \pm 2.0$ | $72.4 \pm 2.8$ | $68.3 \pm 2.3$ | $73.2 \pm 2.1$ |

and a dictionary of $k$ atoms $\theta \in \mathbb{R}^{k \times p}$, a sparse code is defined as a matrix $x^\star(\theta) \in \mathbb{R}^{m \times k}$ that minimizes in $x$ a reconstruction loss $f(x, \theta) := \ell(X_{\texttt{tr}}, x\theta)$ regularized by a sparsity-inducing penalty $g(x)$. Instead of optimizing the dictionary $\theta$ to minimize the reconstruction loss, [60] proposed to optimize an outer problem that depends on the code. Given a set of labels $Y_{\texttt{tr}} \in \{0, 1\}^m$, we consider a logistic regression problem which results in the bilevel optimization problem:

$$\underbrace{\min_{\theta \in \mathbb{R}^{k \times p}, w \in \mathbb{R}^k, b \in \mathbb{R}} \sigma(x^\star(\theta)w + b; y_{\texttt{tr}})}_{\text{outer problem}} \quad \text{subject to} \quad x^\star(\theta) \in \underbrace{\operatorname*{argmin}_{x \in \mathbb{R}^{m \times k}} f(x, \theta) + g(x)}_{\text{inner problem}} . \quad (11)$$

When $\ell$ is the squared Frobenius distance between matrices, and $g$ the elastic net penalty, [60, Eq. 21] derive manually, using optimality conditions (notably the support of the codes selected at the optimum), an explicit re-parameterization of $x^\star(\theta)$ as a linear system involving $\theta$. This closed-form allows for a *direct* computation of the Jacobian of $x^\star$ w.r.t. $\theta$. Similarly, [78] derive first order conditions in the case where $\ell$ is a $\beta$-divergence, while [85] propose to use unrolling of ISTA iterations. Our approach bypasses all of these manual derivations, giving the user more leisure to focus directly on modeling (loss, regularizer) aspects.

We illustrate this on breast cancer survival prediction from gene expression data. We frame it as a binary classification problem to discriminate patients who survive longer than 5 years ($m_1 = 200$) vs patients who die within 5 years of diagnosis ($m_0 = 99$), from $p = 1,000$ gene expression values. As shown in Table 2, solving (11) (Task-driven DictL) reaches a classification performance competitive with state-of-the-art $L_1$ or $L_2$ regularized logistic regression with 100 times fewer variables.

### 4.4 Sensitivity analysis of molecular dynamics

Many physical simulations require solving optimization problems, such as energy minimization in molecular [76] and continuum [9] mechanics, structural optimization [53] and data assimilation [40]. In this experiment, we revisit an example from JAX-MD [76], the problem of finding energy minimizing configurations to a system of $k$ packed particles in a 2-dimensional box of size $\ell$

$$x^\star(\theta) = \operatorname*{argmin}_{x \in \mathbb{R}^{k \times 2}} f(x, \theta) := \sum_{i,j} U(x_{i,j}, \theta),$$

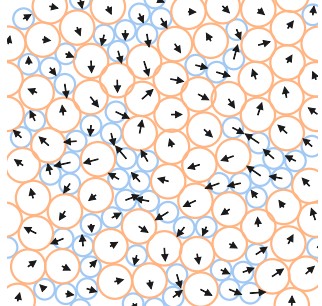

where $x^\star(\theta) \in \mathbb{R}^{k \times 2}$ are the optimal coordinates of the $k$ particles, $U(x_{i,j}, \theta)$ is the pairwise potential energy function, with half the particles at diameter 1 and half at diameter $\theta = 0.6$, which we optimize with a domain-specific optimizer [15]. Here we consider sensitivity of particle position with respect to diameter $\partial x^\star(\theta)$, rather than sensitivity of the total energy from the original experiment. Figure 6 shows results calculated via forward-mode implicit

Figure 6: Particle positions and position sensitivity vectors, with respect to increasing the diameter of the blue particles.

differentiation (JVP). Whereas differentiating the unrolled optimizer happens to work for total energy, here it typically does not even converge, due to the discontinuous optimization method.

## 5 Conclusion

We proposed in this paper an approach for automatic implicit differentiation, allowing the user to freely express the optimality conditions of the optimization problem whose solutions are to be differentiated, directly in Python. The applicability of our approach to a large catalog of optimality conditions is shown in the non-exhaustive list of Table 1, and illustrated by the ease with which we can solve bi-level and sensitivity analysis problems.

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
