# Appendix

## A   More examples of optimality criteria and fixed points

To demonstrate the generality of our approach, we describe in this section more optimality mapping $F$ or fixed point iteration $T$.

**Mirror descent fixed point.**   We again consider the case when $x^\star(\theta)$ is implicitly defined as the solution of (8). We now generalize the projected gradient fixed point beyond Euclidean geometry. Let the Bregman divergence $D_\varphi\colon \mathrm{dom}(\varphi) \times \mathrm{relint}(\mathrm{dom}(\varphi)) \to \mathbb{R}_+$ generated by $\varphi$ be defined by

$$D_\varphi(x, y) \coloneqq \varphi(x) - \varphi(y) - \langle \nabla\varphi(y), x - y \rangle.$$

We define the Bregman projection of $y$ onto $\mathcal{C}(\theta) \subseteq \mathrm{dom}(\varphi)$ by

$$\mathrm{proj}_{\mathcal{C}}^{\varphi}(y, \theta) \coloneqq \underset{x \in \mathcal{C}(\theta)}{\mathrm{argmin}}\, D_\varphi(x, \nabla\varphi^*(y)). \tag{12}$$

Definition (12) includes the mirror map $\nabla\varphi^*(y)$ for convenience. It can be seen as a mapping from $\mathbb{R}^d$ to $\mathrm{dom}(\varphi)$, ensuring that (12) is well-defined. The mirror descent fixed point mapping is then

$$\hat{x} = \nabla\varphi(x)$$
$$y = \hat{x} - \eta\nabla_1 f(x, \theta)$$
$$T(x, \theta) = \mathrm{proj}_{\mathcal{C}}^{\varphi}(y, \theta). \tag{13}$$

Because $T$ involves the composition of several functions, manually deriving its JVP/VJP is error prone. This shows that our approach leveraging autodiff allows to handle more advanced fixed point mappings. A common example of $\varphi$ is $\varphi(x) = \langle x, \log x - \mathbf{1} \rangle$, where $\mathrm{dom}(\varphi) = \mathbb{R}_+^d$. In this case, $D_\varphi$ is the Kullback-Leibler divergence. An advantage of the Kullback-Leibler projection is that it sometimes easier to compute than the Euclidean projection, as we detail in Appendix C.

**Newton fixed point.**   Let $x$ be a root of $G(\cdot, \theta)$, i.e., $G(x, \theta) = 0$. The fixed point iteration of Newton's method for root-finding is

$$T(x, \theta) = x - \eta[\partial_1 G(x, \theta)]^{-1} G(x, \theta).$$

By the chain and product rules, we have

$$\partial_1 T(x, \theta) = I - \eta(...)G(x, \theta) - \eta[\partial_1 G(x, \theta)]^{-1}\partial_1 G(x, \theta) = (1 - \eta)I.$$

Using (3), we get $A = -\partial_1 F(x, \theta) = \eta I$. Similarly,

$$B = \partial_2 T(x, \theta) = \partial_2 F(x, \theta) = -\eta[\partial_1 G(x, \theta)]^{-1}\partial_2 G(x, \theta).$$

Newton's method for optimization is obtained by choosing $G(x, \theta) = \nabla_1 f(x, \theta)$, which gives

$$T(x, \theta) = x - \eta[\nabla_1^2 f(x, \theta)]^{-1}\nabla_1 f(x, \theta). \tag{14}$$

It is easy to check that we recover the same linear system as for the gradient descent fixed point (5). A practical implementation can pre-compute an LU decomposition of $\partial_1 G(x, \theta)$, or a Cholesky decomposition if $\partial_1 G(x, \theta)$ is positive semi-definite.

**Proximal block coordinate descent fixed point.**   We now consider the case when $x^\star(\theta)$ is implicitly defined as the solution

$$x^\star(\theta) \coloneqq \underset{x \in \mathbb{R}^d}{\mathrm{argmin}}\, f(x, \theta) + \sum_{i=1}^{m} g_i(x_i, \theta),$$

where $g_1, \ldots, g_m$ are possibly non-smooth functions operating on subvectors (blocks) $x_1, \ldots, x_m$ of $x$. In this case, we can use for $i \in [m]$ the fixed point

$$x_i = [T(x, \theta)]_i = \mathrm{prox}_{\eta_i g_i}(x_i - \eta_i[\nabla_1 f(x, \theta)]_i, \theta), \tag{15}$$

where $\eta_1, \ldots, \eta_m$ are block-wise step sizes. Clearly, when the step sizes are shared, i.e., $\eta_1 = \cdots = \eta_m = \eta$, this fixed point is equivalent to the proximal gradient fixed point (7) with $g(x, \theta) = \sum_{i=1}^{n} g_i(x_i, \theta)$.

**Quadratic programming.** We now show how to use the KKT conditions discussed in §2.2 to differentiate quadratic programs, recovering Optnet [6] as a special case. To give some intuition, let us start with a simple equality-constrained quadratic program (QP)

$$\operatorname*{argmin}_{z\in\mathbb{R}^p} f(z,\theta) = \frac{1}{2}z^\top Q z + c^\top z \quad \text{subject to} \quad H(z,\theta) = Ez - d = 0,$$

where $Q \in \mathbb{R}^{p\times p}$, $E \in \mathbb{R}^{q\times p}$, $d \in \mathbb{R}^q$. We gather the differentiable parameters as $\theta = (Q, E, c, d)$. The stationarity and primal feasibility conditions give

$$\nabla_1 f(z,\theta) + [\partial_1 H(z,\theta)]^\top \nu = Qz + c + E^\top \nu = 0$$
$$H(z,\theta) = Ez - d = 0.$$

In matrix notation, this can be rewritten as

$$\begin{bmatrix} Q & E^\top \\ E & 0 \end{bmatrix} \begin{bmatrix} z \\ \nu \end{bmatrix} = \begin{bmatrix} -c \\ d \end{bmatrix}. \tag{16}$$

We can write the solution of the linear system (16) as the root $x = (z, \nu)$ of a function $F(x, \theta)$. More generally, the QP can also include inequality constraints

$$\operatorname*{argmin}_{z\in\mathbb{R}^p} f(z,\theta) = \frac{1}{2}z^\top Q z + c^\top z \quad \text{subject to} \quad H(z,\theta) = Ez - d = 0, G(z,\theta) = Mz - h \leq 0.$$

where $M \in \mathbb{R}^{r\times p}$ and $h \in \mathbb{R}^r$. We gather the differentiable parameters as $\theta = (Q, E, M, c, d, h)$. The stationarity, primal feasibility and complementary slackness conditions give

$$\nabla_1 f(z,\theta) + [\partial_1 H(z,\theta)]^\top \nu + [\partial_1 G(z,\theta)]^\top \lambda = Qz + c + E^\top \nu + M^\top \lambda = 0$$
$$H(z,\theta) = Ez - d = 0$$
$$\lambda \circ G(z,\theta) = \operatorname{diag}(\lambda)(Mz - h) = 0$$

In matrix notation, this can be written as

$$\begin{bmatrix} Q & E^\top & M^\top \\ E & 0 & 0 \\ \operatorname{diag}(\lambda)M & 0 & 0 \end{bmatrix} \begin{bmatrix} z \\ \nu \\ \lambda \end{bmatrix} = \begin{bmatrix} -c \\ d \\ \lambda \circ h \end{bmatrix}$$

While $x = (z, \nu, \lambda)$ is no longer the solution of a linear system, it is the root of a function $F(x, \theta)$ and therefore fits our framework. With our framework, no derivation is needed. We simply define $f$, $H$ and $G$ directly in Python.

**Conic programming.** We now show that the differentiation of conic linear programs [3, 5], at the heart of differentiating through cvxpy layers [2], easily fits our framework. Consider the problem

$$z^\star(\lambda), s^\star(\lambda) = \operatorname*{argmin}_{z\in\mathbb{R}^p, s\in\mathbb{R}^m} c^\top z \quad \text{subject to} \quad Ez + s = d, s \in \mathcal{K}, \tag{17}$$

where $\lambda = (c, E, d)$, $E \in \mathbb{R}^{m\times p}$, $d \in \mathbb{R}^m$, $c \in \mathbb{R}^p$ and $\mathcal{K} \subseteq \mathbb{R}^m$ is a cone; $z$ and $s$ are the primal and slack variables, respectively. Every convex optimization problem can be reduced to the form (17). Let us form the skew-symmetric matrix

$$\theta(\lambda) = \begin{bmatrix} 0 & E^\top & c \\ -E & 0 & d \\ -c^\top & -d^\top & 0 \end{bmatrix} \in \mathbb{R}^{N\times N},$$

where $N = p + m + 1$. Following [3, 2, 5], we can use the homogeneous self-dual embedding to reduce the process of solving (17) to finding a root of the residual map

$$F(x,\theta) = \theta \Pi x + \Pi^* x = ((\theta - I)\Pi + I)x, \tag{18}$$

where $\Pi = \operatorname{proj}_{\mathbb{R}^p \times \mathcal{K}^* \times \mathbb{R}_+}$ and $\mathcal{K}^* \subseteq \mathbb{R}^m$ is the dual cone. The splitting conic solver [68], which is based on ADMM, outputs a solution $F(x^\star(\theta), \theta) = 0$ which is decomposed as $x^\star(\theta) = (u^\star(\theta), v^\star(\theta), w^\star(\theta))$. We can then recover the optimal solution of (17) using

$$z^\star(\lambda) = u^\star(\theta(\lambda)) \quad \text{and} \quad s^\star(\lambda) = \operatorname{proj}_{\mathcal{K}^*}(v^\star(\theta(\lambda))) - v^\star(\theta(\lambda)).$$

The key oracle whose JVP/VJP we need is therefore $\Pi$, which is studied in [4]. The projection onto a few cones is available in our library and can be used to express $F$.

**Frank-Wolfe.** We now consider

$$x^\star(\theta) = \operatorname*{argmin}_{x \in \mathcal{C}(\theta) \subset \mathbb{R}^d} f(x, \theta), \tag{19}$$

where $\mathcal{C}(\theta)$ is a convex polytope, i.e., it is the convex hull of vertices $v_1(\theta), \ldots, v_m(\theta)$. The Frank-Wolfe algorithm requires a linear minimization oracle (LMO)

$$s \mapsto \operatorname*{argmin}_{x \in \mathcal{C}(\theta)} \langle s, x \rangle$$

and is a popular algorithm when this LMO is easier to compute than the projection onto $\mathcal{C}(\theta)$. However, since this LMO is piecewise constant, its Jacobian is null almost everywhere. Inspired by SparseMAP [67], which corresponds to the case when $f$ is a quadratic, we rewrite (19) as

$$p^\star(\theta) = \operatorname*{argmin}_{p \in \triangle^m} g(p, \theta) \coloneqq f(V(\theta)p, \theta),$$

where $V(\theta)$ is a $d \times m$ matrix gathering the vertices $v_1(\theta), \ldots, v_m(\theta)$. We then have $x^\star(\theta) = V(\theta)p^\star(\theta)$. Since we have reduced (19) to minimization over the simplex, we can use the projected gradient fixed point to obtain

$$T(p^\star(\theta), \theta) = \operatorname{proj}_{\triangle^m}(p^\star(\theta) - \nabla_1 g(p^*(\theta), \theta)).$$

We can therefore compute the derivatives of $p^\star(\theta)$ by implicit differentiation and the derivatives of $x^\star(\theta)$ by product rule. Frank-Wolfe implementations typically maintain the convex weights of the vertices, which we use to get an approximation of $p^\star(\theta)$. Moreover, it is well-known that after $t$ iterations, at most $t$ vertices are visited. We can leverage this sparsity to solve a smaller linear system. Moreover, in practice, we only need to compute VJPs of $x^\star(\theta)$.

# B Code examples

## B.1 Code examples for optimality conditions

Our library provides several reusable optimality condition mappings $F$ or fixed points $T$. We nevertheless demonstrate the ease of writing some of them from scratch.

**KKT conditions.** As a more advanced example, we now describe how to implement the KKT conditions (6). The stationarity, primal feasibility and complementary slackness conditions read

$$\nabla_1 f(z, \theta_f) + [\partial_1 G(z, \theta_G)]^\top \lambda + [\partial_1 H(z, \theta_H)]^\top \nu = 0$$
$$H(z, \theta_H) = 0$$
$$\lambda \circ G(z, \theta_G) = 0.$$

Using `jax.vjp` to compute vector-Jacobian products, this can be implemented as

```python
grad = jax.grad(f)

def F(x, theta):
  z, nu, lambd = x
  theta_f, theta_H, theta_G = theta

  _, H_vjp = jax.vjp(H, z, theta_H)
  stationarity = (grad(z, theta_f) + H_vjp(nu)[0])

  primal_feasability = H(z, theta_H)

  _, G_vjp = jax.vjp(G, z, theta_G)
  stationarity += G_vjp(lambd)[0]
  comp_slackness = G(z, theta_G) * lambd

  return stationarity, primal_feasability, comp_slackness
```

Figure 7: KKT conditions $F(x, \theta)$

Similar mappings $F$ can be written if the optimization problem contains only equality constraints or only inequality constraints.

**Mirror descent fixed point.**   Letting $\eta = 1$ and denoting $\theta = (\theta_f, \theta_{\mathrm{proj}})$, the fixed point (13) is

$$\hat{x} = \nabla\varphi(x)$$
$$y = \hat{x} - \nabla_1 f(x, \theta_f)$$
$$T(x, \theta) = \mathrm{proj}_{\mathcal{C}}^{\varphi}(y, \theta_{\mathrm{proj}}).$$

We can then implement it as follows.

```
grad = jax.grad(f)

def T(x, theta):
  theta_f, theta_proj = params
  x_hat = phi_mapping(x)
  y = x_hat - grad(x, theta_f)
  return bregman_projection(y, theta_proj)
```

Figure 8: Mirror descent fixed point $T(x, \theta)$

Although not considered in this example, the mapping $\nabla\varphi$ could also depend on $\theta$ if necessary.

## B.2   Code examples for experiments

We now sketch how to implement our experiments using our framework. In the following, `jnp` is short for `jax.numpy`. In all experiments, we only show how to compute gradients with the outer objective. We can then use these gradients with gradient-based solvers to solve the outer objective.

**Multiclass SVM experiment.**

```
X_tr, Y_tr, X_val, Y_val = load_data()

def W(x, theta):  # dual-primal map
  return jnp.dot(X_tr.T, Y_tr - x) / theta

def f(x, theta):  # inner objective
  return (0.5 * theta * jnp.sum(W(x, theta) ** 2) +
          jnp.vdot(x, Y_tr))

grad = jax.grad(f)
proj = jax.vmap(projection_simplex)  # row-wise projections
def T(x, theta):
  return proj(x - grad(x, theta))

@custom_fixed_point(T)
def msvm_dual_solver(init_x, theta):
  # [...]
  return x_star  # solution of the dual objective

def outer_loss(lambd):
  theta = jnp.exp(lambd)
  x_star = msvm_dual_solver(init_x, theta)  # inner solution
  Y_pred = jnp.dot(W(x_star, theta), X_val)
  return 0.5 * jnp.sum((Y_pred - Y_val) ** 2)

print(jax.grad(outer_loss)(lambd))
```

Figure 9: Code example for the multiclass SVM experiment.

**Task-driven dictionary learning experiment.**

```python
X_tr, y_tr = load_data()

def f(x, theta):  # dictionary loss
  residual = X_tr - jnp.dot(x, theta)
  return huber_loss(residual)

grad = jax.grad(f)
def T(x, theta):  # proximal gradient fixed point
  return prox_lasso(x - grad(x, theta))

@custom_fixed_point(T)
def sparse_coding(init_x, theta):  # inner objective
  # [...]
  return x_star  # lasso solution

def outer_loss(theta, w):  # task-driven loss
  x_star = sparse_coding(init_x, theta)  # sparse codes
  y_pred = jnp.dot(x_star, w)
  return logloss(y_tr, y_pred)

print(jax.grad(outer_loss, argnums=(0,1)))
```

Figure 10: Code example for the task-driven dictionary learning experiment.

**Dataset distillation experiment.**

```python
X_tr, y_tr = load_data()

logloss = jax.vmap(loss.multiclass_logistic_loss)

def f(x, theta, l2reg=1e-3):  # inner objective
  scores = jnp.dot(theta, x)
  distilled_labels = jnp.arange(10)
  penalty = l2reg * jnp.sum(x * x)
  return jnp.mean(logloss(distilled_labels, scores)) + penalty

F = jax.grad(f)

@custom_root(F)
def logreg_solver(init_x, theta):
  # [...]
  return x_star

def outer_loss(theta):
  x_star = logreg_solver(init_x, theta)  # inner solution
  scores = jnp.dot(X_tr, x_star)
  return jnp.mean(logloss(y_tr, scores))

print(jax.grad(outer_loss)(theta))
```

Figure 11: Code example for the dataset distillation experiment.

**Molecular dynamics experiment.**

```
energy_fn = soft_sphere_energy_fn(diameter)
init_fn, apply_fn = jax_md.minimize.fire_descent(
    energy_fn, shift_fn)

x0 = random.uniform(key, (N, 2))
R0 = L * x0  # transform to physical coordinates
R = lax.fori_loop(
    0, num_optimization_steps,
    body_fun=lambda t, state: apply_fn(state, t=t),
    init_val=init_fn(R0)).position
x_star = R / L

def F(x, diameter):  # normalized forces
  energy_fn = soft_sphere_energy_fn(diameter)
  normalized_energy_fn = lambda x: energy_fn(L * x)
  return -jax.grad(normalized_energy_fn)(x)

dx = root_jvp(F, x_star, diameter, 1.0,
              solve=linear_solve.solve_bicgstab)

print(dx)
```

Figure 12: Code for the molecular dynamics experiment.

## C  Jacobian products

Our library provides numerous reusable building blocks. We describe in this section how to compute their Jacobian products. As a general guideline, whenever a projection enjoys a closed form, we leave the Jacobian product to the autodiff system.

### C.1  Jacobian products of projections

We describe in this section how to compute the Jacobian products of the projections (in the Euclidean and KL senses) onto various convex sets. When the convex set does not depend on any variable, we simply denote it $\mathcal{C}$ instead of $\mathcal{C}(\theta)$.

**Non-negative orthant.**  When $\mathcal{C}$ is the non-negative orthant, $\mathcal{C} = \mathbb{R}^d_+$, we obtain $\text{proj}_{\mathcal{C}}(y) = \max(y, 0)$, where the maximum is evaluated element-wise. This is also known as the ReLu function. The projection in the KL sense reduces to the exponential function, $\text{proj}^{\varphi}_{\mathcal{C}}(y) = \exp(y)$.

**Box constraints.**  When $\mathcal{C}(\theta)$ is the box constraints $\mathcal{C}(\theta) = [\theta_1, \theta_2]^d$ with $\theta \in \mathbb{R}^2$, we obtain

$$\text{proj}_{\mathcal{C}}(y, \theta) = \text{clip}(y, \theta_1, \theta_2) := \max(\min(y, \theta_2), \theta_1).$$

This is trivially extended to support different boxes for each coordinate, in which case $\theta \in \mathbb{R}^{d \times 2}$.

**Probability simplex.**  When $\mathcal{C}$ is the standard probability simplex, $\mathcal{C} = \triangle^d$, there is no analytical solution for $\text{proj}_{\mathcal{C}}(y)$. Nevertheless, the projection can be computed exactly in $O(d)$ expected time or $O(d \log d)$ worst-case time [22, 63, 33, 26]. The Jacobian is given by $\text{diag}(s) - ss^\top/\|s\|_1$, where $s \in \{0, 1\}^d$ is a vector indicating the support of $\text{proj}_{\mathcal{C}}(y)$ [62]. The projection in the KL sense, on the other hand, enjoys a closed form: it reduces to the usual softmax $\text{proj}^{\varphi}_{\mathcal{C}}(y) = \exp(y)/\sum_{j=1}^{d} \exp(y_j)$.

**Box sections.** Consider now the Euclidean projection $z^\star(\theta) = \text{proj}_{\mathcal{C}}(y, \theta)$ onto the set $\mathcal{C}(\theta) = \{z \in \mathbb{R}^d \colon \alpha_i \leq z_i \leq \beta_i, i \in [d]; w^\top z = c\}$, where $\theta = (\alpha, \beta, w, c)$. This projection is a singly-constrained bounded quadratic program. It is easy to check (see, e.g., [66]) that an optimal solution satisfies for all $i \in [d]$

$$z_i^\star(\theta) = [L(x^\star(\theta), \theta)]_i := \text{clip}(w_i x^\star(\theta) + y_i, \alpha_i, \beta_i)$$

where $L \colon \mathbb{R} \times \mathbb{R}^n \to \mathbb{R}^d$ is the dual-primal mapping and $x^\star(\theta) \in \mathbb{R}$ is the optimal dual variable of the linear constraint, which should be the root of

$$F(x^\star(\theta), \theta) = L(x^\star(\theta), \theta)^\top w - c.$$

The root can be found, e.g., by bisection. The gradient $\nabla x^\star(\theta)$ is given by $\nabla x^\star(\theta) = B^\top/A$ and the Jacobian $\partial z^\star(\theta)$ is obtained by application of the chain rule on $L$.

**Norm balls.** When $\mathcal{C}(\theta) = \{x \in \mathbb{R}^d \colon \|x\| \leq \theta\}$, where $\|\cdot\|$ is a norm and $\theta \in \mathbb{R}_+$, $\text{proj}_{\mathcal{C}}(y, \theta)$ becomes the projection onto a norm ball. The projection onto the $\ell_1$-ball reduces to a projection onto the simplex, see, e.g., [33]. The projections onto the $\ell_2$ and $\ell_\infty$ balls enjoy a closed-form, see, e.g., [69, §6.5]. Since they rely on simple composition of functions, all three projections can therefore be automatically differentiated.

**Affine sets.** When $\mathcal{C}(\theta) = \{x \in \mathbb{R}^d \colon Ax = b\}$, where $A \in \mathbb{R}^{p \times d}$, $b \in \mathbb{R}^p$ and $\theta = (A, b)$, we get

$$\text{proj}_{\mathcal{C}}(y, \theta) = y - A^\dagger(Ay - b) = y - A^\top(AA^\top)^{-1}(Ay - b)$$

where $A^\dagger$ is the Moore-Penrose pseudoinverse of $A$. The second equality holds if $p < d$ and $A$ is full rank. A practical implementation can pre-compute a factorization of the Gram matrix $AA^\top$. Alternatively, we can also use the KKT conditions.

**Hyperplanes and half spaces.** When $\mathcal{C}(\theta) = \{x \in \mathbb{R}^d \colon a^\top x = b\}$, where $a \in \mathbb{R}^d$ and $b \in \mathbb{R}$ and $\theta = (a, b)$, we get

$$\text{proj}_{\mathcal{C}}(y, \theta) = y - \frac{a^\top y - b}{\|a\|_2^2} a.$$

When $\mathcal{C}(\theta) = \{x \in \mathbb{R}^d \colon a^\top x \leq b\}$, we simply replace $a^\top y - b$ in the numerator by $\max(a^\top y - b, 0)$.

**Transportation and Birkhoff polytopes.** When $\mathcal{C}(\theta) = \{X \in \mathbb{R}^{p \times d} \colon X\mathbf{1}_d = \theta_1, X^\top \mathbf{1}_p = \theta_2, X \geq 0\}$, the so-called transportation polytope, where $\theta_1 \in \triangle^p$ and $\theta_2 \in \triangle^d$ are marginals, we can compute approximately the projections, both in the Euclidean and KL senses, by switching to the dual or semi-dual [17]. Since both are unconstrained optimization problems, we can compute their Jacobian product by implicit differentiation using the gradient descent fixed point. An advantage of the KL geometry here is that we can use Sinkhorn [28], which is a GPU-friendly algorithm. The Birkhoff polytope, the set of doubly stochastic matrices, is obtained by fixing $\theta_1 = \theta_2 = \mathbf{1}_d/d$.

**Order simplex.** When $\mathcal{C}(\theta) = \{x \in \mathbb{R}^d \colon \theta_1 \geq x_1 \geq x_2 \geq \cdots \geq x_d \geq \theta_2\}$, a so-called order simplex [49, 16], the projection operations, both in the Euclidean and KL sense, reduce to isotonic optimization [58] and can be solved exactly in $O(d \log d)$ time using the Pool Adjacent Violators algorithm [14]. The Jacobian of the projections and efficient product with it are derived in [31, 18].

**Polyhedra.** More generally, we can consider polyhedra, i.e., sets of the form $\mathcal{C}(\theta) = \{x \in \mathbb{R}^d \colon Ax = b, Cx \leq d\}$, where $A \in \mathbb{R}^{p \times d}$, $b \in \mathbb{R}^p$, $C \in \mathbb{R}^{m \times d}$, and $d \in \mathbb{R}^m$. There are several ways to differentiate this projection. The first is to use the KKT conditions as detailed in §2.2. A second way is consider the dual of the projection instead, which is the maximization of a quadratic function subject to **non-negative constraints** [69, §6.2]. That is, we can reduce the projection on a polyhedron to a problem of the form (8) with non-negative constraints, which we can in turn implicitly differentiate easily using the projected gradient fixed point, combined with the projection on the non-negative orthant. Finally, we apply the dual-primal mapping , which enjoys a closed form and is therefore amenable to autodiff, to obtain the primal projection.

## C.2 Jacobian products of proximity operators

We provide several proximity operators, including for the lasso (soft thresholding), elastic net and group lasso (block soft thresholding). All satisfy closed form expressions and can be differentiated automatically via autodiff. For more advanced proximity operators, which do not enjoy a closed form, recent works have derived their Jacobians. The Jacobians of fused lasso and OSCAR were derived in [65]. For general total variation, the Jacobians were derived in [80, 24].

## D Jacobian precision proofs

*Proof of Theorem 1.* To simplify notations, we note $A_\star := A(x^\star, \theta)$ and $\hat{A} := A(\hat{x}, \theta)$, and similarly for $B$ and $J$. We have by definition of the Jacobian estimate function $A_\star J_\star = B_\star$ and $\hat{A}\hat{J} = \hat{B}$. Therefore we have

$$J(\hat{x}, \theta) - \partial x^\star(\theta) = \hat{A}^{-1}\hat{B} - A_\star^{-1}B_\star$$
$$= \hat{A}^{-1}\hat{B} - \hat{A}^{-1}B_\star + \hat{A}^{-1}B_\star - A_\star^{-1}B_\star$$
$$= \hat{A}^{-1}(\hat{B} - B_\star) + (\hat{A}^{-1} - A_\star^{-1})B_\star.$$

For any invertible matrices $M_1, M_2$, it holds that $M_1^{-1} - M_2^{-1} = M_1^{-1}(M_2 - M_1)M_2^{-1}$, so

$$\|M_2^{-1} - M_2^{-1}\|_{\mathrm{op}} \leq \|M_1^{-1}\|_{\mathrm{op}}\|M_2 - M_1\|_{\mathrm{op}}\|M_2^{-1}\|_{\mathrm{op}}.$$

Therefore,

$$\|\hat{A}^{-1} - A_\star^{-1}\|_{\mathrm{op}} \leq \frac{1}{\alpha^2}\|\hat{A} - A_\star\|_{\mathrm{op}} \leq \frac{\gamma}{\alpha^2}\|\hat{x} - x^\star(\theta)\|.$$

As a consequence, the second term in $J(\hat{x}, \theta) - \partial x^\star(\theta)$ can be upper bounded and we obtain

$$\|J(\hat{x}, \theta) - \partial x^\star(\theta)\| \leq \|\hat{A}^{-1}(\hat{B} - B_\star)\| + \|(\hat{A}^{-1} - A_\star^{-1})B_\star\|$$
$$\leq \|\hat{A}^{-1}\|_{\mathrm{op}}\|\hat{B} - B_\star\| + \frac{\gamma}{\alpha^2}\|\hat{x} - x^\star(\theta)\| \|B_\star\|,$$

which yields the desired result. $\qquad\square$

**Corollary 1** (Jacobian precision for gradient descent fixed point). *Let $f$ be such that $f(\cdot, \theta)$ is twice differentiable and $\alpha$-strongly convex and $\nabla_1^2 f(\cdot, \theta)$ is $\gamma$-Lipschitz (in the operator norm) and $\partial_2\nabla_1 f(x, \theta)$ is $\beta$-Lipschitz and bounded in norm by $R$. The estimated Jacobian evaluated at $\hat{x}$ is then given by*
$$J(\hat{x}, \theta) = -(\nabla_1^2 f(\hat{x}, \theta))^{-1}\partial_2\nabla_1 f(\hat{x}, \theta).$$
*For all $\theta \in \mathbb{R}^n$, and any $\hat{x}$ estimating $x^\star(\theta)$, we have the following bound for the approximation error of the estimated Jacobian*

$$\|J(\hat{x}, \theta) - \partial x^\star(\theta)\| \leq \left(\frac{\beta}{\alpha} + \frac{\gamma R}{\alpha^2}\right)\|\hat{x} - x^\star(\theta)\|.$$

*Proof of Corollary 1.* This follows from Theorem 1, applied to this specific $A(x, \theta)$ and $B(x, \theta)$. $\quad\square$

For proximal gradient descent, where $T(x, \theta) = \mathrm{prox}_{\eta g}(x - \eta\nabla_1 f(x, \theta), \theta)$, this yields

$$A(x, \theta) = I - \partial_1 T(x, \theta) = I - \partial_1\mathrm{prox}_{\eta g}(x - \eta\nabla_1 f(x, \theta), \theta)(I - \eta\nabla_1^2 f(x, \theta))$$
$$B(x, \theta) = \partial_2\mathrm{prox}_{\eta g}(x - \eta\nabla_1 f(x, \theta), \theta) - \eta\partial_1\mathrm{prox}_{\eta g}(x - \eta\nabla_1 f(x, \theta), \theta)\partial_2\nabla_1 f(x, \theta).$$

We now focus in the case of proximal gradient descent on an objective $f(x, \theta) + g(x)$, where $g$ is smooth and does not depend on $\theta$. This is the case in our experiments in §4.3. Recent work also exploits local smoothness of solutions to derive similar bounds [13, Theorem 13]

**Corollary 2** (Jacobian precision for proximal gradient descent fixed point). *Let $f$ be such that $f(\cdot, \theta)$ is twice differentiable and $\alpha$-strongly convex and $\nabla_1^2 f(\cdot, \theta)$ is $\gamma$-Lipschitz (in the operator norm) and $\partial_2\nabla_1 f(x, \theta)$ is $\beta$-Lipschitz and bounded in norm by $R$. Let $g : \mathbb{R}^d \to \mathbb{R}$ be a twice-differentiable $\mu$-strongly convex (with special case $\mu = 0$ being only convex), for which the function*

$\Gamma_\eta(x, \theta) = \nabla^2 g(\text{prox}_{\eta g}(x - \eta \nabla_1 f(x, \theta))$ *is $\kappa_\eta$-Lipschitz in it first argument. The estimated Jacobian evaluated at $\hat{x}$ is then given by*

$$J(\hat{x}, \theta) = -(\nabla_1^2 f(\hat{x}, \theta) + \Gamma_\eta(\hat{x}, \theta))^{-1} \partial_2 \nabla_1 f(\hat{x}, \theta) \,.$$

*For all $\theta \in \mathbb{R}^n$, and any $\hat{x}$ estimating $x^\star(\theta)$, we have the following bound for the approximation error of the estimated Jacobian*

$$\|J(\hat{x}, \theta) - \partial x^\star(\theta)\| \leq \left( \frac{\beta + \kappa_\eta}{\alpha + \mu} + \frac{\gamma R}{(\alpha + \mu)^2} \right) \|\hat{x} - x^\star(\theta)\| \,.$$

*Proof of Corollary 2.* First, let us note that $\text{prox}_{\eta g}(y, \theta)$ does not depend on $\theta$, since $g$ itself does not depend on $\theta$, and is therefore equal to classical proximity operator of $\eta g$ which, with a slight overload of notations, we denote as $\text{prox}_{\eta g}(y)$ (with a single argument). In other words,

$$\begin{cases} \text{prox}_{\eta g}(y, \theta) & = \text{prox}_{\eta g}(y) \,, \\ \partial_1 \text{prox}_{\eta g}(y, \theta) & = \partial \text{prox}_{\eta g}(y) \,, \\ \partial_2 \text{prox}_{\eta g}(y, \theta) & = 0 \,. \end{cases}$$

Regarding the first claim (expression of the estimated Jacobian evaluated at $\hat{x}$), we first have that $\text{prox}_{\eta g}(y)$ is the solution to $(x' - y) + \eta \nabla g(x') = 0$ in $x'$ - by first-order condition for a smooth convex function. We therefore have that

$$\text{prox}_{\eta g}(y) = (I + \eta \nabla g)^{-1}(y)$$
$$\partial \text{prox}_{\eta g}(y) = (I_d + \eta \nabla^2 g(\text{prox}_{\eta g}(y)))^{-1} \,,$$

the first $I$ and inverse being functional identity and inverse, and the second $I_d$ and inverse being in the matrix sense, by inverse rule for Jacobians $\partial h(z) = [\partial h^{-1}(h(z))]^{-1}$ (applied to the prox).

As a consequence, we have, for $\Gamma_\eta(x, \theta) = \nabla^2 g(\text{prox}_{\eta g}(x - \eta \nabla_1 f(x, \theta))$ that

$$\begin{aligned} A(x, \theta) &= I_d - (I_d + \eta \Gamma_\eta(x, \theta))^{-1}(I_d - \eta \nabla_1^2 f(x, \theta)) \\ &= (I_d + \eta \Gamma_\eta(x, \theta))^{-1}[I_d + \eta \Gamma_\eta(x, \theta) - (I_d - \eta \nabla_1^2 f(x, \theta))] \\ &= \eta(I_d + \eta \Gamma_\eta(x, \theta))^{-1}(\nabla_1^2 f(x, \theta) + \Gamma_\eta(x, \theta)) \\ B(x, \theta) &= -\eta(I_d + \eta \Gamma_\eta(x, \theta))^{-1} \partial_2 \nabla_1 f(x, \theta) \,. \end{aligned}$$

As a consequence, for all $x \in \mathbb{R}^d$, we have that

$$J(x, \theta) = -(\nabla_1^2 f(x, \theta) + \Gamma_\eta(x, \theta))^{-1} \partial_2 \nabla_1 f(x, \theta) \,.$$

In the following, we modify slightly the notation of both $A$ and $B$, writing

$$\begin{aligned} \tilde{A}(x, \theta) &= \nabla_1^2 f(x, \theta) + \Gamma_\eta(x, \theta) \\ \tilde{B}(x, \theta) &= -\partial_2 \nabla_1 f(x, \theta) \,. \end{aligned}$$

With the current hypotheses, following along the proof of Theorem 1, we have that $\tilde{A}$ is $(\alpha + \mu)$ well-conditioned, and $(\gamma + \kappa_\eta)$-Lipschitz in its first argument, and $\tilde{B}$ is $\beta$-Lipschitz in its first argument and bounded in norm by $R$. The same reasoning yields

$$\|J(\hat{x}, \theta) - \partial x^\star(\theta)\| \leq \left( \frac{\beta + \kappa_\eta}{\alpha + \mu} + \frac{\gamma R}{(\alpha + \mu)^2} \right) \|\hat{x} - x^\star(\theta)\| \,.$$

$\square$

# E  The Lasso case

Our approach to differentiate the solution of a root equation $F(x, \theta) = 0$ is valid as long as the smooth implicit function theorem holds, namely, as long as $F$ continuously differentiable near a solution $(x_0, \theta_0)$ and $\nabla_1 F(x_0, \theta_0)$ is invertible. While the first assumption is easy to check when $F$ is continuously differentiable *everywhere*, it does not always hold when this is not the case, e.g.,

when $F$ involves the proximity operator of a non-smooth function. In such cases, one may therefore have to study theoretically the properties of the function $F$ near the solutions $(x(\theta), \theta)$ to justify differentiation using the smooth implicit function theorem. Here, we develop such an analysis to justify the use of our approach to differentiate the solution of a Lasso regression problem with respect to the regularization parameter. We note that the smooth implicit function theorem has already been used for this problem [12, 13]; here we justify why it is a valid approach, even though $F$ itself is not continuously differentiable everywhere. More precisely, we consider the Lasso problem:

$$\forall \theta \in \mathbb{R}, \quad x^\star(\theta) = \operatorname*{argmin}_{x \in \mathbb{R}^d} \frac{1}{2} \|\Phi x - b\|_2^2 + e^\theta \|x\|_1, \tag{20}$$

where $\Phi \in \mathbb{R}^{m \times d}$ and $b \in \mathbb{R}^m$. Note that we parameterize the regularization parameter as $e^\theta$ to ensure that it remains strictly positive for $\theta \in \mathbb{R}$, but the analysis below does not depend on this particular choice. This is a typical non-smooth optimization problem where we may want to use a proximal gradient fixed point equation (9) to differentiate the solution. In this case we have $f(x, \theta) = (1/2)\|\Phi x - b\|_2^2$, hence $\nabla_1 f(x, \theta) = \Phi^\top (\Phi x - b)$, and $g(x, \theta) = e^\theta \|x\|_1$, hence $\operatorname{prox}_{\eta g}(y, \theta) = \mathrm{ST}(y, \eta e^\theta)$ where $\mathrm{ST} : \mathbb{R}^d \times \mathbb{R} \to \mathbb{R}^d$ is the soft-thresholding operator: $\mathrm{ST}(a, b)_i = \operatorname{sign}(a_i) \times \max(|a_i| - b, 0)$. The root equation that characterizes the solution is therefore, for any $\eta > 0$:

$$F_\eta(x, \theta) = x - \mathrm{ST}\left(x - \eta \Phi^\top(\Phi x - b), \eta e^\theta\right) = 0. \tag{21}$$

It is well-known that, under mild assumptions, (20) has a unique solution for any $\theta \in \mathbb{R}$, and that the solution path $\{x^\star(\theta) : \theta \in \mathbb{R}\}$ is continuous and piecewise linear, with a finite number of non-differentiable points (often called "kinks") [79, 61]. Since the function $\theta \mapsto x^\star(\theta)$ is not differentiable at the kinks, the smooth implicit function theorem using (21) is not applicable at those points (as any other methods to compute the Jacobian of $x^\star(\theta)$, such as unrolling). Interestingly, though, the following result shows that, under mild assumptions, the smooth implicit function theorem using (21) is valid on *all* other points of the solution path, thus justifying the use of our approach to compute the Jacobian of $x^\star(\theta)$ whenever it exists. Note that we state this theorem under some assumption on the matrix $\Phi$ that is sufficient to ensure that the solution to the lasso is unique [79, Lemma 4], but that weaker assumptions could be possible (see proof).

**Theorem 2.** *If the entries of $\Phi$ are drawn from a continuous probability distribution on $\mathbb{R}^{m \times d}$, then the smooth implicit function theorem using the root equation* (21) *holds at any point $(x^\star(\theta), \theta)$ that is not a kink of the solution path, with probability one.*

*Proof.* We first show that $F_\eta$ is continuously differentiable in a neighborhood of $(x^\star(\theta), \theta)$, for any $\theta$ that is not a kink. Since the ST operator is continuously differentiable everywhere except on the closed set:

$$\mathcal{S} = \left\{(a, b) \in \mathbb{R}^d \times \mathbb{R} \ : \ \exists i \in [1, d], |a_i| = b\right\},$$

it suffices to show from (21) that $(x^\star(\theta) - \eta \Phi^\top(\Phi x^\star(\theta) - b), \eta e^\theta) \notin \mathcal{S}$. For that purpose, we characterize $x^\star(\theta)$ using the subgradient of (20) as follows: there exists $\gamma \in \mathbb{R}^d$ such that

$$\Phi^\top(b - \Phi x^\star(\theta)) = e^\theta \gamma,$$

where

$$\begin{cases} \gamma_i = \operatorname{sign}(x^\star(\theta)_i) & \text{if } x^\star(\theta)_i \neq 0, \\ \gamma_i \in [-1, 1] & \text{otherwise.} \end{cases}$$

We thus need to show that $(x^\star(\theta) + \eta e^\theta \gamma, \eta e^\theta) \notin \mathcal{S}$. We first see that, for any $i \in [1, d]$ such that $x^\star(\theta)_i \neq 0$,

$$|x^\star(\theta)_i + \eta e^\theta \gamma_i| = |x^\star(\theta)_i + \eta e^\theta \operatorname{sign}(x^\star(\theta)_i)| > \eta e^\theta.$$

The case $x^\star(\theta)_i \neq 0$ requires more care, since the property $\gamma_i \in [-1, 1]$ is not sufficient to show that $|x^\star(\theta)_i + \eta e^\theta \gamma_i| = \eta e^\theta |\gamma_i|$ is not equal to $\eta e^\theta$: we need to show that, in fact, $|\gamma_i| < 1$. For that purpose, let $\mathcal{E}(\theta) = \{i \in [1, d] \ : \ |\gamma_i| = 1\}$. Denoting $\Phi_{\mathcal{E}(\theta)}$ the matrix made of the columns of $\Phi$ in $\mathcal{E}(\theta)$, we know that, under the assumptions of Theorem 2, with probability one the matrix $\Phi_{\mathcal{E}(\theta)}^\top \Phi_{\mathcal{E}(\theta)}$ is invertible and the lasso problem has a unique solution given by $x^\star(\theta)_{\mathcal{E}(\theta)^C} = 0$ and

$$x^\star(\theta)_{\mathcal{E}(\theta)} = (\Phi_{\mathcal{E}(\theta)}^\top \Phi_{\mathcal{E}(\theta)})^{-1} \left(\Phi_{\mathcal{E}(\theta)}^\top b - e^\theta s(\theta)_{\mathcal{E}(\theta)}\right), \tag{22}$$

where $s(\theta) = \text{sign}(\Phi^\top(b - \Phi x^\star(\theta)))) \in \{-1, 0, 1\}^d$ [79]. Furthermore, we know that $\mathcal{E}(\theta)$ is constant between two successive kinks [61], so if $x^\star(\theta)$ is not a kink then there is a neighborhood $[\theta_1, \theta_2]$ of $\theta$ such as $\mathcal{E}(\theta') = \mathcal{E}(\theta)$ and $s(\theta')_{\mathcal{E}(\theta')} = s(\theta)_{\mathcal{E}(\theta)}$, for any $\theta' \in [\theta_1, \theta_2]$. Let us now assume that $\Phi$ is such that for any $\mathcal{E} \subset [1, d]$ and $s \in \{-1, 1\}^{|\mathcal{E}|}$, $\Phi_{\mathcal{E}}^\top \Phi_{\mathcal{E}}$ is invertible and $(\Phi_{\mathcal{E}}^\top \Phi_{\mathcal{E}})^{-1} s$ has no coordinate equal to zero. This happens with probability one under the assumptions of Theorem 2 since the set of singular matrices is measure zero. Then we see from (22) that, for $\theta' \in [\theta_1, \theta_2]$ and $i \in \mathcal{E}(\theta)$, $x^\star(\theta')_i$ is an affine and non-constant function of $e^{\theta'}$. Since in addition $x^\star(\theta_1)_i$ and $x^\star(\theta_2)_i$ are either both nonnegative or nonpositive, then necessarily $x^\star(\theta)_i$ is positive or negative, respectively. In other words, we have shown that $|\gamma_i| = 1 \implies x^\star(\theta)_i \neq 0$, or equivalently that $x^\star(\theta)_i = 0 \implies |\gamma_i| < 1$. From this we deduce that for any $i \in [1, d]$ such that $x^\star(\theta)_i = 0$,

$$|x^\star(\theta)_i + \eta e^\theta \gamma_i| = \eta e^\theta |\gamma_i| < \eta e^\theta .$$

This concludes the proof that $(x^\star(\theta) + \eta e^\theta \gamma, \eta e^\theta) \notin \mathcal{S}$, and therefore that $F_\eta$ is continuously differentiable in a neighborhood of $(x^\star(\theta), \theta)$. The second condition for the smooth implicit theorem to hold, namely, the invertibility of $\nabla_1 F_\eta(x^\star(\theta), \theta)$, is easily obtained by explicit computation [12, 13, Proposition 1]  □

## F   Experimental setup and additional results

Our experiments use JAX [21], which is Apache2-licensed and scikit-learn [71], which is BSD-licensed.

### F.1   Hyperparameter optimization of multiclass SVMs

**Experimental setup.** Synthetic datasets were generated using scikit-learn's `sklearn.datasets.make_classification` [71], following a model adapted from [50]. All datasets consist of $m = 700$ training samples belonging to $k = 5$ distinct classes. To simulate problems of different sizes, the number of features is varied as $p \in \{100, 250, 500, 750, 1000, 2000, 3000, 4000, 5000, 7500, 10000\}$, with 10% of features being informative and the rest random noise. In all cases, an additional $m_{\text{val}} = 200$ validation samples were generated from the same model to define the outer problem.

For the inner problem, we employed three different solvers: (i) mirror descent, (ii) (accelerated) proximal gradient descent and (iii) block coordinate descent. Hyperparameters for all solvers were individually tuned manually to ensure convergence across the range of problem sizes. For mirror descent, a stepsize of 1.0 was used for the first 100 steps, following a inverse square root decay afterwards up to a total of 2500 steps. For proximal gradient descent, a stepsize of $5 \cdot 10^{-4}$ was used for 2500 steps. The block coordinate descent solver was run for 500 iterations. All solvers used the same initialization, namely, $x_{\text{init}} = \frac{1}{k} 1_{m \times k}$, which satisfies the dual constraints.

For the outer problem, gradient descent was used with a stepsize of $5 \cdot 10^{-3}$ for the first 100 steps, following a inverse square root decay afterwards up to a total of 150 steps.

Conjugate gradient was used to solve the linear systems in implicit differentiation for at most 2500 iterations.

All results reported pertaining CPU runtimes were obtained using an internal compute cluster. GPU results were obtained using a single NVIDIA P100 GPU with 16GB of memory per dataset. For each dataset size, we report the average runtime of an individual iteration in the outer problem, alongside a 90% confidence interval estimated from the corresponding 150 runtime values.

**Additional results**   Figure 13 compares the runtime of implicit differentiation and unrolling on GPU. These results highlight a fundamental limitation of the unrolling approach in memory-limited systems such as accelerators, as the inner solver suffered from out-of-memory errors for most problem sizes ($p \geq 2000$ for mirror descent, $p \geq 750$ for proximal gradient and block coordinate descent). While it might be possible to ameliorate this limitation by reducing the maximum number of iterations in the inner solver, doing so might lead to additional challenges [84] and require careful tuning.

Figure 14 depicts the validation loss (value of the outer problem objective function) at convergence. It shows that all approaches were able to solve the outer problem, with solutions produced by different

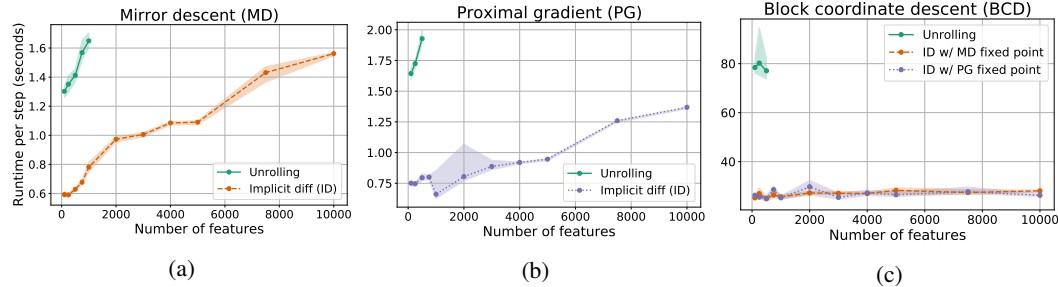

(a)

(b)

(c)

Figure 13: GPU runtime comparison of implicit differentiation and unrolling for hyperparameter optimization of multiclass SVMs for multiple problem sizes (same setting as Figure 4). Error bars represent 90% confidence intervals. Absent data points were due to out-of-memory errors (16 GB maximum).

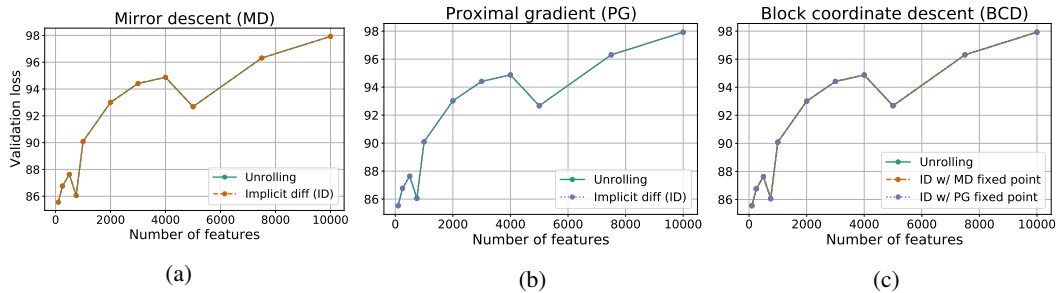

(a)

(b)

(c)

Figure 14: Value of the outer problem objective function (validation loss) for hyperparameter optimization of multiclass SVMs for multiple problem sizes (same setting as Figure 4). As can be seen, all methods performed similarly in terms of validation loss. This confirms that the faster runtimes for implicit differentiation compared to unrolling shown in Figure 4 (CPU) and Figure 13 (GPU) are not at the cost of worse validation loss.

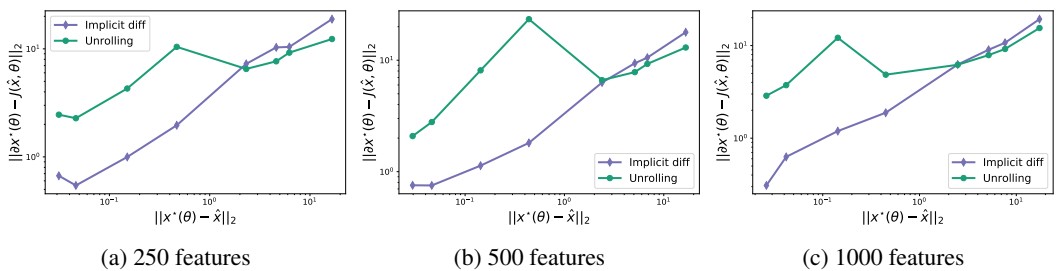

(a) 250 features

(b) 500 features

(c) 1000 features

Figure 15: Jacobian error $\|\partial x^\star(\theta) - J(\hat{x}, \theta)\|_2$ (see also Definition 1) evaluated with a regularization parameter of $\theta = 1$, as a function of solution error $\|x^\star(\theta) - \hat{x}\|_2$ when varying the number of features, on the multiclass SVM task (see Appendix F.1 for a detailed description of the experimental setup). The ground-truth solution $x^\star(\theta)$ is computed using the liblinear solver [37] available in scikit-learn [71] with a very low tolerance of $10^{-9}$. Unlike in Figure 3, which was on ridge regression, the ground-truth Jacobian $\partial x^\star(\theta)$ cannot be computed in closed form, in the more difficult setting of multiclass SVMs. We therefore use a finite difference to approximately compute $\partial x^\star(\theta)$. Our results nevertheless confirm similar trends as in Figure 3.

approaches being qualitatively indistinguishable from each other across the range of problem sizes considered.

Figure 15 shows the Jacobian error achieved as a function of the solution error, when varying the number of features.

## F.2 Task-driven dictionary learning

We downloaded from http://acgt.cs.tau.ac.il/multi_omic_benchmark/download.html a set of breast cancer gene expression data together with survival information generated by the TCGA Research Network (https://www.cancer.gov/tcga) and processed as explained by [74]. The gene expression matrix contains the expression value for p=20,531 genes in m=1,212 samples, from which we keep only the primary tumors (m=1,093). From the survival information, we select the patients who survived at least five years after diagnosis ($m_1 = 200$), and the patients who died before five years ($m_0 = 99$), resulting in a cohort of $m = 299$ patients with gene expression and binary label. Note that non-selected patients are those who are marked as alive but were not followed for 5 years.

To evaluate different binary classification methods on this cohort, we repeated 10 times a random split of the full cohort into a training (60%), validation (20%) and test (20%) sets. For each split and each method, 1) the method is trained with different parameters on the training set, 2) the parameter that maximizes the classification AUC on the validation set is selected, 3) the method is then re-trained on the union of the training and validation sets with the selected parameter, and 4) we measure the AUC of that model on the test set. We then report, for each method, the mean test AUC over the 10 repeats, together with a 95% confidence interval defined a mean $\pm$ 1.96 $\times$ standard error of the mean.

We used Scikit Learn's implementation of logistic regression regularized by $\ell_1$ (lasso) and $\ell_2$ (ridge) penalty from sklearn.linear_model.LogisticRegression, and varied the C regularization parameter over a grid of 10 values: $\{10^{-5}, 10^{-3}, \ldots, 10^4\}$. For the unsupervised dictionary learning experiment method, we estimated a dictionary from the gene expression data in the training and validation sets, using sklearn.decomposition.DictionaryLearning(n_components=10, alpha=2.0), which produces sparse codes in $k = 10$ dimensions with roughly 50% nonzero coefficients by minimizing the squared Frobenius reconstruction distance with lasso regularization on the code. We then use sklearn.linear_model.LogisticRegression to train a logistic regression on the codes, varying the ridge regularization parameter C over a grid of 10 values $\{10^{-1}, 10^0, \ldots, 10^8\}$.

Finally, we implemented the task-driven dictionary learning model (11) with our toolbox, following the pseudo-code in Figure 10. Like for the unsupervised dictionary learning experiment, we set the dimension of the codes to $k = 10$, and a fixed elastic net regularization on the inner optimization problem to ensure that the codes have roughly 50% sparsity. For the outer optimization problem, we solve an $\ell_2$ regularized ridge regression problem, varying again the ridge regularization parameter C over a grid of 10 values $\{10^{-1}, 10^0, \ldots, 10^8\}$. Because the outer problem is non-convex, we minimize it using the Adam optimizer [56] with default parameters.

## F.3 Dataset Distillation

**Experimental setup.** For the inner problem, we used gradient descent with backtracking line-search, while for the outer problem we used gradient descent with momentum and a fixed step-size. The momentum parameter was set to $0.9$ while the step-size was set to $1$.

Figure 5 was produced after 4000 iterations of the outer loop on CPU (Intel(R) Xeon(R) Platinum P-8136 CPU @ 2.00GHz), which took 1h55. Unrolled differentiation took instead 8h:05 (4 times more) to run the same number of iterations. As can be seen in Figure 16, the output is the same in both approaches.

Dataset Distillation (MNIST). Generalization Accuracy: 0.8556

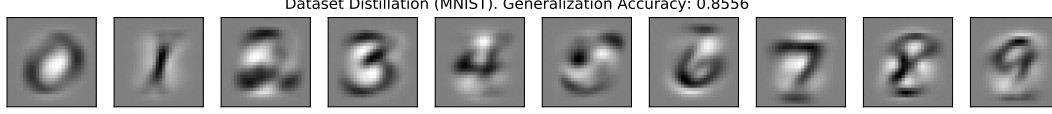

Figure 16: Distilled MNIST dataset $\theta \in \mathbb{R}^{k \times p}$ obtained by solving (10) through unrolled differentiation. Although there is no qualitative difference, the implicit differentiation approach is 4 times faster.

### F.4 Molecular dynamics

Our experimental setup is adapted from the JAX-MD example notebook available at `https://github.com/google/jax-md/blob/master/notebooks/meta_optimization.ipynb`.

We emphasize that calculating the gradient of the total energy objective, $f(x, \theta) = \sum_{ij} U(x_{i,j}, \theta)$, with respect to the diameter $\theta$ of the smaller particles, $\nabla_1 f(x, \theta)$, does not require implicit differentiation or unrolling. This is because $\nabla_1 f(x, \theta) = 0$ at $x = x^\star(\theta)$:

$$\nabla_\theta f(x^\star(\theta), \theta) = \partial x^\star(\theta)^\top \nabla_1 f(x^\star(\theta), \theta) + \nabla_2 f(x^\star(\theta), \theta) = \nabla_2 f(x^\star(\theta), \theta).$$

This is known as Danskin's theorem or envelope theorem. Thus instead, we consider sensitivities of position $\partial x^\star(\theta)$ directly, which does require implicit differentiation or unrolling.

Our results comparing implicit and unrolled differentiation for calculating the sensitivity of position are shown in Figure 17. We use BiCGSTAB [81] to perform the tangent linear solve. Like in the original JAX-MD experiment, we use $k = 128$ particles in $m = 2$ dimensions.

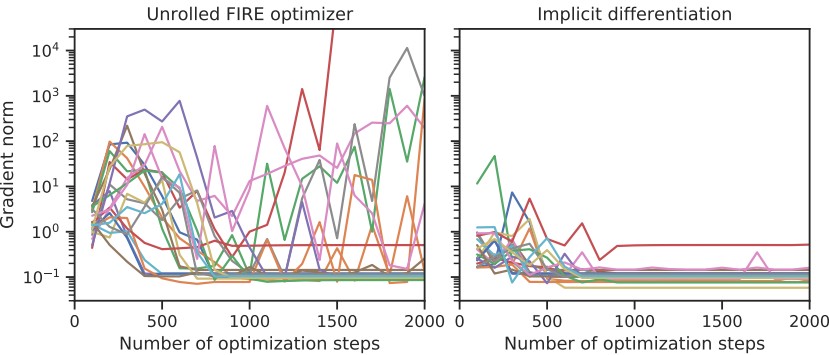

Figure 17: L1 norm of position sensitivities in the molecular dynamics simulations, for 40 different random initial conditions (different colored lines). Gradients through the unrolled FIRE optimizer [15] for many initial conditions do not converge, in contrast to implicit differentiation.