# OpenReview forum: "Efficient and Modular Implicit Differentiation"
_NeurIPS.cc/2022/Conference — NeurIPS 2022 Accept_

### Official Review · Reviewer_Mi39 · 2022-06-30

**Rating:** 6
**Confidence:** 4
**Soundness:** 3 good
**Presentation:** 4 excellent
**Contribution:** 3 good

**Summary:**

This paper introduces a Jax package for automatic implicit differentiation. Specifically, the authors propose an efficient and modular approach for implicit differentiation of optimization problems. Their approach combines the benefits of autodiff and implicit differentiation. The authors also provide precision guarantees of the proposed approach. Moreover, the author demonstrate the effectiveness of the proposed approach in bi-level optimization and the sensitivity analysis.

**Questions:**

Please see weakness 2 and 3. If the authors can address these two concerns, I will improve my scores.

**Limitations:**

Yes.

**Strengths And Weaknesses:**

My detailed comments are given as below.

Strength:

1 The motivation of this paper is clear. I believe that this paper makes an important contribution to very relevant topics, e.g., the bi-level optimization, implicit deep neural networks.

2 This paper is well written and the proposed approach is easy to follow. The authors provide some illustrative figures to demonstrate their approach. I feel this is good.

3 The authors implemented four illustrative examples to demonstrate the effectiveness of their approach.

Weakness:

1 The main contribution of this paper is about the software, but the theoretical contribution is overstated. The proof of the theorem is quite standard and I do not get some new insight from it.

2 Direct runtime comparisons with existing methods are missing. The proposed approach is based on implicit differentiation which usually requires additional computational costs. Thus, the direct runtime comparison is necessary to demonstrate the efficiency of the proposed approach.

3 Recently, implicit deep learning has attracted many attentions, which is very relevant to the topic of this paper. An implementation example of implicit deep neural networks should be included. Moreover,  many Jacobian-free methods e.g., [1-3] have been proposed to reduce the computational cost. The comparisons (runtime and accuracy) with these methods are preferred.

[1] Fung, Samy Wu, et al. "Fixed point networks: Implicit depth models with Jacobian-free backprop." (2021).

[2] Geng, Zhengyang, et al. "On training implicit models." Advances in Neural Information Processing Systems 34 (2021): 24247-24260.

[3] Ramzi, Zaccharie, et al. "SHINE: SHaring the INverse Estimate from the forward pass for bi-level optimization and implicit models." arXiv preprint arXiv:2106.00553 (2021).

---

> ### Author Response · Authors · 2022-08-01
> **Runtime comparison already addressed, other comments taken into account in the revision**
>
> We thank the reviewer for the positive review and constructive comments. We believe that the runtime comparison request is already addressed in the paper. We have addressed your other comments below and in the revised manuscript.
>
> > The main contribution of this paper is about the software, but the theoretical contribution is overstated. The proof of the theorem is quite standard and I do not get some new insight from it.
>
> We provide a general result that is easier to apply in our context than [Higham 2002, Theorem 7.2]. It is used to provide a theoretical explanation for the phenomenon shown in Figure 3, when comparing the precision of implicit differentiation and unrolling, which is insightful and novel to our knowledge. Simplicity of the proof and insight are in our opinion not incompatible. We feel that the description of this experimental phenomenon would not be complete without this theoretical insight.
>
> > Direct runtime comparisons with existing methods are missing. The proposed approach is based on implicit differentiation which usually requires additional computational costs. Thus, the direct runtime comparison is necessary to demonstrate the efficiency of the proposed approach.
>
> Figure 4 in the paper already shows a runtime comparison of implicit differentiation vs. unrolling on CPU for 3 algorithms (mirror descent, proximal gradient and block coordinate descent) and Figure 13 in the Appendix shows the same comparison on GPU.
>
> > Recently, implicit deep learning has attracted many attentions, which is very relevant to the topic of this paper. An implementation example of implicit deep neural networks should be included. Moreover, many Jacobian-free methods e.g., [1-3] have been proposed to reduce the computational cost.
>
> Our software contains an example of deep equilibrium network (DEQ) with Anderson acceleration. We added the suggested 3 references to the revised manuscript.

---

> > ### Comment · Reviewer_Mi39 · 2022-08-08
> > **Re**
> >
> > The author's response addressed most of my concerns. I tend to accept this paper.

---

### Official Review · Reviewer_SAnA · 2022-07-09

**Rating:** 9
**Confidence:** 3
**Soundness:** 4 excellent
**Presentation:** 4 excellent
**Contribution:** 4 excellent

**Summary:**

Implicit differentiation can be used to through the solution to an optimization problem without having to backpropagate through the method by which this solution was determined. While this approach has been used in many different contexts, implicit differentiation must often be tailored towards the specific problem in question. This article introduces a library that implements automatic implicit differentiation, addressing this limitation. It goes on to provide precision estimates if an optimization problem has not been exactly solved, and concludes with several examples showcasing the library.

**Questions:**

- Returning to the dataset distillation example, reference [55] only uses a few gradient steps for the inner loop to make their algorithm scalable. Do the authors think their approach towards dataset distillation for logistic regression would scale towards deeper networks?
- l. 224: What do the authors mean by "implicit differentiation gains a factor of t compared to automatic differentiation"? Moreover, does automatic differentiation in this context refer to automatic differentiation through the t iterations rather than implicit automatic differentiation?
- l. 227: What is missing to apply these results to hypergradients? Would that simply be a matter of applying the chain rule to compute the gradients with respect to the objective and obtain corresponding precision guarantees?
- I did not understand the illustration in figure 6; since $\theta\in\mathbb{R}^k$ specifies the diameter of each particle, isn’t $\partial x^{\ast}(\theta)\in\mathbb{R}^{k\times(k\times 2)}$? Is the figure depicting the diagonal elements of that Jacobian?
- Where is digit ‘9’ in figure 5?

One typo: l. 324: 'converge, due to the'

**Limitations:**

I believe the authors have adequately addressed the limitations of their work.

**Strengths And Weaknesses:**

This submission fills an important gap by providing a library implementing automatic implicit differentiation. In my view, this promises to make the use of implicitly defined layers (regardless of their specific form) more accessible for practitioners, in particular making it easier to try out a particular idea. The library implementation is well documented and seems to integrate well with other Jax packages. The article itself is well written, providing a nice balance of motivation, code, theory, and examples. Theorem 1 is closely related to existing results about inverse stability (as the authors note), but it is helpful to have it stated explicitly in the context of implicit differentiation. The proof seems to be correct.

As a minor issue, I was surprised to not see any example applications in the context of deep networks. Reference [55] implements dataset distillation on deep networks, yet the authors seem to be focused on a logistic regression case (see Questions). The code provided in the supplementary material also seems to include deep learning examples, but the authors make no note of this in the appendix. I don't see the inclusion of deep learning examples in the article as a necessity at all, but was a little surprised about the lack thereof.

---

> ### Author Response · Authors · 2022-08-01
> **Thank you for the extremely positive feedback**
>
> We thank the reviewer for the extremely positive feedback and constructive comments.
>
> > As a minor issue, I was surprised to not see any example applications in the context of deep networks. Reference [55] implements dataset distillation on deep networks, yet the authors seem to be focused on a logistic regression case (see Questions). The code provided in the supplementary material also seems to include deep learning examples, but the authors make no note of this in the appendix. I don't see the inclusion of deep learning examples in the article as a necessity at all, but was a little surprised about the lack thereof.
>
> >Returning to the dataset distillation example, reference [55] only uses a few gradient steps for the inner loop to make their algorithm scalable. Do the authors think their approach towards dataset distillation for logistic regression would scale towards deeper networks?
>
> In this particular example, we strove for simplicity: we wanted to show that it was possible to implement a dataset distillation example in less than 100 lines of code (the current example counts 67 lines counting comments).
>
> As shown in the other deep learning examples, the implicit differentiation mechanism can scale to objectives with a deep network, and so it would be possible to extend this example to use a deep network instead of a linear one, with a modest increase in runtime and code complexity.
>
> > l. 224: What do the authors mean by "implicit differentiation gains a factor of t compared to automatic differentiation"? Moreover, does automatic differentiation in this context refer to automatic differentiation through the t iterations rather than implicit automatic differentiation?
>
> Our remark about the “gains a factor of t” refers to the observation that we prove in Theorem 1 that the error in Jacobian estimate using the implicit differentiation is upper bounded (up to a multiplicative constant) by the error in the solution estimate, while [1, Proposition 3.2] shows that when the Jacobian is instead estimated by unrolling, then the error in the Jacobian estimate is upper bounded (up to a multiplicative constant) by the error in the solution estimate multiplied by t, the number of iterations performed. Hence our upper bound for the Jacobian estimation by implicit differentiation  is, up to the constants, better than the known upper bounds for Jacobian estimation by unrolling by a factor of t. Regarding the second question, yes, automatic differentiation refers to automatic differentiation through the t iterations.
>
> > I did not understand the illustration in figure 6; since θ∈Rk specifies the diameter of each particle, isn’t ∂x∗(θ)∈Rk×(k×2)? Is the figure depicting the diagonal elements of that Jacobian?
>
> θ is the diameter of the blue particles and is therefore in R, not R^k (we do not assume that each individual particle has its own diameter). Therefore, x*(θ) is a function from R to R^{2k}, i.e., it outputs the 2-dimensional coordinates of the k particles at equilibrium, for a given diameter θ. The Jacobian is then a vector of size 2k, which gives the 2-dimensional coordinates of the k particles. The diameter of the orange particles is fixed to 1.
>
> > Where is digit ‘9’ in figure 5?
>
> We just wanted a 3 x 3 figure for saving space. The full example can be generated from the code in the supplementary material /source_code/examples/implicit_diff/plot_dataset_distillation.py
>
> > One typo: l. 324: 'converge, due to the'
>
> Fixed, thank you!

---

### Official Review · Reviewer_sKqm · 2022-07-11

**Rating:** 6
**Confidence:** 5
**Soundness:** 3 good
**Presentation:** 4 excellent
**Contribution:** 2 fair

**Summary:**

EDIT: After the discussion with the authors and the revisions they submitted, I modified my overall score from 3 to 6, the presentation score from 3 to 4, and the soundness score from 1 to 3.

The paper presents a blueprint for automatic implicit differentiation of solutions to optimization problems, along with its implementation in the JAX library. They claim that their blueprint is widely applicable by listing several common optimization problem templates (e.g., those solvable by mirror descent, proximal gradient descent, Newton's method, conic programming, etc) that they claim one can apply their blueprint to. The main idea of the blueprint is to find a fixed point equation representing the optimality conditions and then apply the implicit function theorem to this fixed point equation. They also prove a theorem regarding the precision of the estimated implicit Jacobian in terms of the precision of the estimated fixed point, which they give numerical support for. Finally, they report numerical experiments for four problems, primarily comparing their method to unrolling.

**Questions:**

In line 262, what is meant by unrolling if the algorithm is nonsmooth?

**Limitations:**

A major limitation of the work is that it does not apply to the nonsmooth setting in which the fixed point equation associated to the optimality conditions is not continuously differentiable. Many of the optimization problems coming from machine learning are nonsmooth and so this severely limits the impact of the proposed blueprint.

**Strengths And Weaknesses:**

One strength of the paper is the software implementation in JAX, which is user-friendly, designed to be broadly applicable, and seemingly efficient; later theoretical flaws limit the justified use of this software and thus make this contribution less significant.

The paper is ambiguous when defining functions, not specifying if a function is continuous, differentiable, twice differentiable, etc, at key points, like in the blueprint and the theorem about Jacobian precision. For example, around line 98 when differentiating the root is introduced as the main principle which the rest of the paper relies on, there is no specification on the regularity of the function F besides that it should be "a user-provided mapping, capturing the optimality conditions of a problem." However, the argument that follows relies on the smooth implicit function theorem, which in this context requires the function F to be C1 (continuously differentiable) in a neighborhood. This continues in line 118 where the chain rule is used without specifying that F is smooth, in line 168 when the regularity of f, G, and H are not specified, in line 199 for the definition of Jacobian estimate, in line 210 for the main/only theorem, etc.

This ambiguity becomes important for applying the results. Many of the functions involved in Table 1 on line 161, for instance, are frequently nonsmooth in machine learning contexts (e.g., the prox, the projection operator, etc), and thus the results developed in the paper are not applicable to them. There is mention in line 183 that, because the prox is differentiable a.e., that the smooth implicit function theorem can be applied a.e.. This is false - stronger assumptions than differentiability at a point are needed to apply the smooth implicit function theorem, i.e., F must be C1 on an open neighborhood of the solution (along with an invertibility condition). While nonsmooth implicit function theorems exist, e.g., [Clarke 1990], [Bolte et al 2022], they are nowhere mentioned and their requirements to be applied are different than those of the smooth implicit function since the generalized gradients involved are set-valued. This issue comes up again in line 202 where it's assumed that the solution xstar is differentiable, which is not true in general (e.g., the Lasso solution can have kinks as a function of the l1 weight).

Some numerical experiments are also affected by this, for example the euclidean projection onto the simplex isn't smooth (line 257) and cannot be treated by the blueprint. While it's true that sometimes a smooth fixed point equation can be associated to a nonsmooth problem, such as in section 4.1 where the mirror descent formulation was used with the smooth KL projection on the simplex, there is no discussion of this phenomenon in the paper, nor a way to systematically construct such smooth fixed equations in general. The nonsmoothness also raises questions for the numerical comparisons regarding unrolling, which is mentioned in the questions section.

There is also a concern about the originality of the ideas in the paper. The main idea of the blueprint, to associate an equation modeling optimality to the problem instead of using the set-valued optimality conditions, is trivial in the smooth case since the optimality conditions will no longer be set-valued. Since little beyond line 183 is said about implicitly differentiating nonsmooth functions, the contribution of the blueprint here reduces to the fact that we can associate a fixed point equation to optimality conditions, which is well-known, e.g., for maximal monotone inclusions in [Bauschke, Combettes 2011].

Because the results are ultimately relegated to the smooth setting, I find the paper to be lacking in originality and significance. The main result that remains is the Jacobian precision theorem for smooth functions which, while compelling, is not sufficient on its own nor in conjunction the numerical experiments for smooth problems to deserve publication here. I think an integration of the nonsmooth case is necessary to make the contribution significant enough to worthy of publication, especially due to the prevalence of nonsmooth optimization problems in machine learning.

"Optimization and nonsmooth analysis" - FH Clarke 1990

"Nonsmooth Implicit Differentiation for Machine Learning and Optimization" - Jérôme Bolte, Tam Le, Edouard Pauwels, Antonio Silveti-Falls 2021

"Convex analysis and monotone operator theory in Hilbert spaces" - HH Bauschke, PL Combettes, 2011

---

> ### Author Response · Authors · 2022-08-01
> **Assumptions clarified, "current limitations" paragraph added**
>
> > The paper does not always specify if a function is continuous, differentiable, twice differentiable
>
> We agree that it’s better to make the assumptions clear rather than implying that the assumptions of the implicit function theorem that we invoke apply everywhere. We clarified the assumptions everywhere in the manuscript (modifications are highlighted in blue color and assumptions already present before submission are highlighted in olive color).
>
> > For ex, around l98 there is no specification on the regularity of the function F
>
> We believe the reviewer missed the sentence l.100-102, where we clarify the regularity conditions that the function F must satisfy for the smooth implicit function theorem to hold, namely, a ““continuously differentiable F with invertible Jacobian”.
>
> > Many of the functions involved in Table 1 are nonsmooth [...]
>
> First we would like to clarify that we did not write “because the prox if differentiable a.e., the smooth implicit function theorem can be applied a.e.”, which would indeed have been wrong. We simply mentioned that the prox operator is a.e. differentiable to remind the reader that differentiability of the prox operator is rather frequent, even for non-smooth optimization problems. But we agree that it can induce confusion and decided to disambiguate the conditions where the implicit function theorem holds, by adding that our framework applies if, in addition to being a.e. differentiable, the prox is continuously differentiable in a neighborhood of the solution and the invertibility condition holds (l.186-188).
>
> > While nonsmooth implicit function theorems exist, e.g., [Clarke 1990], [Bolte et al 2022]
>
> As explained above, our focus is on situations where the (smooth) implicit function theorem holds. We clarified the assumptions for this in Definition 1 (l. 202) and in Theorem 1 (l.213). In addition, we added a sentence in the new “Current limitations” paragraph to mention that extending the approach by using a nonsmooth implicit function theorem is an interesting future work, citing the references [Clarke 1990] and [Bolte et al 2021].
>
> > The euclidean projection onto the simplex isn't smooth (line 257)
>
> We agree with the reviewer that the projection is not differentiable everywhere, but would like to clarify that it is the case almost everywhere. This follows from the fact that a projection is the gradient of a smooth function (with Lipschitz gradients) and by Rademacher’s theorem. In fact, as shown in Appendix C (l.635), the Jacobian exists almost everywhere and is piecewise constant. In practice, this implies our approach is valid and provides the correct derivative of the objective function for almost all values of theta, which we find sufficient to optimize numerically in theta since in practice we never encounter a point of non-differentiability.
>
> > The nonsmoothness also raises questions for the numerical comparisons regarding unrolling
>
> As mentioned in the paper, projection and proximal operators are differentiable a.e., therefore so is unrolling of the projected / proximal gradient algorithm.
>
> > In line 262, what is meant by unrolling if the algorithm is nonsmooth?
>
> We backpropagate through the computational graph generated by the algorithm, including proximal and projection operators (which are differentiable a.e.).
>
> > The results are ultimately relegated to the smooth setting
>
> There seems to be a fundamental disagreement: the reviewer thinks the paper limitations are blocking publication while we (as well as other reviewers and many readers) think the paper is valuable despite them: it presents a reduction to differentiating roots / fixed points, easy-to-use software in JAX, a large variety of experimental results and new Jacobian precision guarantees.
>
> We also emphasize that implicit differentiation of the solution of non-smooth optimization problems is a very recent field of research. The recent reference of Bolte et al develops some new theory but it’s not clear if it has a practical impact. Likewise, it is only recently that backpropagation on differentiable almost everywhere functions has been theoretically investigated (again, by Bolte et al). Yet, ReLus have been routinely used in deep learning pipelines for years.
>
> Overall, we believe that the reviewer misinterpreted the positioning of our paper. We do not claim that we tackle the theory of nonsmooth implicit differentiation, and do not think that we should be evaluated on this ground. We acknowledge that there is currently a gap between theory and practice. However, we still think that this paper nevertheless provides a worthy contribution to practitioners, with some simple theoretical results when they apply; and hope the reviewer will not block its publication.
>
> We have clarified assumptions everywhere in the paper and have acknowledged limitations in an explicit paragraph at the end of Section 2. We hope that this will convince the reviewer to increase their score.

---

> > ### Comment · Reviewer_sKqm · 2022-08-06
> > **Tractability of this assumption?**
> >
> > >We simply mentioned that the prox operator is a.e. differentiable to remind the reader that differentiability of the prox operator is rather frequent, even for non-smooth optimization problems...
> >
> > I agree that the additional assumption that the prox is continuously differentiable on a neighborhood of the solution addresses the issue of applying the smooth implicit function theorem. The catch is in ensuring that this assumption holds, since being differentiable almost everywhere is not a sufficient condition for this. It's not obvious if one can guarantee this assumption will hold unless the prox is simply differentiable everywhere.
> >
> > >We agree with the reviewer that the projection is not differentiable everywhere, but would like to clarify that it is the case almost everywhere. This follows from the fact that a projection is the gradient of a smooth function (with Lipschitz gradients) and by Rademacher’s theorem. In fact, as shown in Appendix C (l.635), the Jacobian exists almost everywhere and is piecewise constant. In practice, this implies our approach is valid and provides the correct derivative of the objective function for almost all values of theta, which we find sufficient to optimize numerically in theta since in practice we never encounter a point of non-differentiability.
> >
> > I disagree that a piecewise constant Jacobian is sufficient. Because of the assumption you have added, to do bilevel optimization with your method it's necessary to ensure that every point visited by the algorithm admits a neighborhood on which the function F is C1. This is not guaranteed by almost everywhere differentiability nor a piecewise constant Jacobian. This speaks to the difficulty of trying to ensure the additional assumption in a bilevel optimization setting.
> >
> > >Overall, we believe that the reviewer misinterpreted the positioning of our paper. We do not claim that we tackle the theory of nonsmooth implicit differentiation, and do not think that we should be evaluated on this ground. We acknowledge that there is currently a gap between theory and practice. However, we still think that this paper nevertheless provides a worthy contribution to practitioners, with some simple theoretical results when they apply; and hope the reviewer will not block its publication.
> >
> > I understand that you do not claim to tackle the theory of nonsmooth implicit differentiation. However, in the blueprint and the applications, you are using implicit differentation on nonsmooth functions which is where the issue comes from. Regarding the contribution to practitioners, I agree that the software package is of a high quality. Yet, if I am a practitioner trying to solve a bilevel optimization problem, how can I know if my problem fits into your famework if I cannot verify this additional assumption holds at all necessary points?
> >
> > Small note: In the current limitations sections it's written that the approach applies when x* is differentiable at theta but it should be written that the approach applies when x* is differentiable in a neighborhood of theta since, under the assumptions that are now in the paper, the solution x* will always be differentiable in a neighborhood of theta by the smooth implicit function theorem.

---

> > > ### Author Response · Authors · 2022-08-08
> > > **Practical details**
> > >
> > > >Yet, if I am a practitioner trying to solve a bilevel optimization problem, how can I know if my problem fits into your famework if I cannot verify this additional assumption holds at all necessary points?
> > >
> > > Checking that the implicit function theorem assumptions hold in theory can indeed be challenging, but is often not necessary in practice, as there is often a metric one cares about. For bilevel optimization, a natural way for a practitioner to check if our framework works is to check that the outer objective value is decreasing.
> > >
> > > All the root objectives and fixed points mentioned in the paper (gradient descent, projected gradient, proximal gradient, mirror descent, KKT, …) are implemented in the library and have been tested successfully either through experiments in the paper or examples in the library.
> > >
> > > It would be possible to detect when the matrix A is singular and to issue a warning to the user if this happens. We chose not to as, again, we have not observed any issues in practice.
> > >
> > > We emphasize once more that it is unclear whether the theory of Bolte et al will lead to any practical algorithmic improvement compared to what we are already doing. Many successful methodologies in ML have theoretical guarantees in restricted theoretical settings but are applied more broadly in practice.
> > >
> > > > Small note: In the current limitations sections it's written that the approach applies when x* is differentiable at theta but it should be written that the approach applies when x* is differentiable in a neighborhood of theta since, under the assumptions that are now in the paper, the solution x* will always be differentiable in a neighborhood of theta by the smooth implicit function theorem.
> > >
> > > We agree with you. We removed “x* is differentiable at theta” and now simply write “we note that the approach developed in this section theoretically only applies to settings where the implicit function theorem is valid, namely, where optimality conditions satisfy the differentiability and invertibility conditions stated in Section 2.1”.

---

> > > > ### Comment · Reviewer_sKqm · 2022-08-09
> > > > **Score raised**
> > > >
> > > > I still contend that the additional assumption fixes the theoretical soundness at the expense of applicability since the assumption is so strict. Realistically, the blueprint can only be used for bilevel optimization with prox operators that are everywhere differentiable (since otherwise it will be challenging/impossible to verify that all points visited by the algorithm admit a neighborhood that avoids the nonsmooth points; almost everywhere differentiability does not fix this). To be very clear, this means nonsmooth problems like the lasso are **not** actually covered by the blueprint presented.
> > > >
> > > > That being said, many papers include a mix of examples which do and do not fall into the scope of theoretical analyses, the authors are very right in this regard. The soundness has been improved by adding the assumption and the clarity has been improved by the revisions + adding the current limitations paragraph, which more clearly outlines the boundary of what's presented (except for the lasso statement, which is false). I will raise my overall score and the soundness and presentation scores. I remain the contribution score because the paper is still very limited in scope and the scope is effectively the same after the assumption (smooth prox operators).

---

> > > > > ### Author Response · Authors · 2022-08-09
> > > > > **The Lasso case**
> > > > >
> > > > > > I still contend that the additional assumption fixes the theoretical soundness at the expense of applicability since the assumption is so strict. Realistically, the blueprint can only be used for bilevel optimization with prox operators that are everywhere differentiable (since otherwise it will be challenging/impossible to verify that all points visited by the algorithm admit a neighborhood that avoids the nonsmooth points; almost everywhere differentiability does not fix this). To be very clear, this means nonsmooth problems like the lasso are not actually covered by the blueprint presented.
> > > > >
> > > > > > That being said, many papers include a mix of examples which do and do not fall into the scope of theoretical analyses, the authors are very right in this regard. The soundness has been improved by adding the assumption and the clarity has been improved by the revisions + adding the current limitations paragraph, which more clearly outlines the boundary of what's presented (except for the lasso statement, which is false). I will raise my overall score and the soundness and presentation scores. I remain the contribution score because the paper is still very limited in scope and the scope is effectively the same after the assumption (smooth prox operators).
> > > > >
> > > > > We thank the reviewer for acknowledging the modifications we have made and for raising their score.
> > > > >
> > > > > While we agree that the hypothesis of the smooth implicit function theorem may be challenging to check for general nonsmooth optimization problems, we would like to clarify that they hold at least for lasso regression, under mild hypothesis over the design matrix. To support this claim, we added Appendix E with a formal statement and proof that the Jacobian of the lasso solution with respect to the regularization parameter can be computed with the implicit function theorem wherever it is differentiable (i.e., everywhere except on a finite number of “kinks”), for general design matrices. We believe this new result clarifies that our setting encompasses more than bilevel optimization with smooth prox operators, and would like to thank the reviewer for their comments that led us to this new result.

---

> > > > > > ### Comment · Reviewer_sKqm · 2022-08-09
> > > > > > **Reply to lasso case**
> > > > > >
> > > > > > >While we agree that the hypothesis of the smooth implicit function theorem may be challenging to check for general nonsmooth optimization problems, we would like to clarify that they hold at least for lasso regression, under mild hypothesis over the design matrix. To support this claim, we added Appendix E with a formal statement and proof that the Jacobian of the lasso solution with respect to the regularization parameter can be computed with the implicit function theorem wherever it is differentiable (i.e., everywhere except on a finite number of “kinks”), for general design matrices. We believe this new result clarifies that our setting encompasses more than bilevel optimization with smooth prox operators, and would like to thank the reviewer for their comments that led us to this new result.
> > > > > >
> > > > > > I stress that these challenging conditions must be checked for all points visited by the algorithm during bilevel optimization (which could change depending on the algorithm used to optimize the outer loss, hyperparameters of that algorithm, initialization, etc). The new result does not fix this since you can still land on a kink during training. Unless there is a way to prove that this doesn't happen, saying the blueprint applies is misleading - the assumptions of the blueprint are not guaranteed to be met during training.
> > > > > >
> > > > > > Besides this, when its written "nonsmooth problems like the lasso" (or similar) in the paper I interpret this to mean fixed point equations with function F which are almost everywhere differentiable but not everywhere differentiable, or more concretely for the sake of example: prox operators that are not everywhere differentiable. In this class of functions, there are for instance prox operators that have sets of nondifferentiable points that are dense (combine the "Remarque" on page 176 of [Zahorski 1946] with proposition 2.3 in [Combettes et all 2019]). For these functions, it can be shown that the blueprint doesn't work even for a single point. The smooth implicit function theorem can never be applied here, despite being a prox operator, almost everywhere differentiable, etc.
> > > > > >
> > > > > >
> > > > > > "Sur l’ensemble des points de non-dérivabilité d’une fonction continue" - Zygmunt Zahorski 1946
> > > > > >
> > > > > > "Deep Neural Network Structures Solving Variational Inequalities" - Patrick L. Combettes and Jean-Christophe Pesquet 2019

---

### Meta-Review · Area_Chair_FmJz · 2022-08-21

**Recommendation:** Accept
**Confidence:** Certain

**Metareview:**

The reviewers have discussed the paper at length and have reached a consensus after the authors have clarified the applicability and limitations of their proposed method. I recommend that the authors continue to polish their manuscript with the points they raised in their summary to the Area Chairs and congratulate them on the acceptance of their submission.

**Award:**

No

---

### Decision · Program_Chairs · 2022-09-14

Accept